# Efficient Robust Bayesian Optimization for Arbitrary Uncertain inputs

**Lin Yang**
Huawei Noah's Ark Lab
China
yanglin33@huawei.com

**Junlong Lyu**
Huawei Noah's Ark Lab
Hong Kong SAR, China
lyujunlong@huawei.com

**Wenlong Lyu**
Huawei Noah's Ark Lab
China
lvwenlong2@huawei.com

**Zhitang Chen**
Huawei Noah's Ark Lab
Hong Kong SAR, China
chenzhitang2@huawei.com

## Abstract

Bayesian Optimization (BO) is a sample-efficient optimization algorithm widely employed across various applications. In some challenging BO tasks, input uncertainty arises due to the inevitable randomness in the optimization process, such as machining errors, execution noise, or contextual variability. This uncertainty deviates the input from the intended value before evaluation, resulting in significant performance fluctuations in final result. In this paper, we introduce a novel robust Bayesian Optimization algorithm, AIRBO, which can effectively identify a robust optimum that performs consistently well under arbitrary input uncertainty. Our method directly models the uncertain inputs of arbitrary distributions by empowering the Gaussian Process with the Maximum Mean Discrepancy (MMD) and further accelerates the posterior inference via Nyström approximation. Rigorous theoretical regret bound is established under MMD estimation error and extensive experiments on synthetic functions and real problems demonstrate that our approach can handle various input uncertainties and achieve a state-of-the-art performance.

## 1 Introduction

Bayesian Optimization (BO) is a powerful sequential decision-making algorithm for high-cost black-box optimization. Owing to its remarkable sample efficiency and capacity to balance exploration and exploitation, BO has been successfully applied in diverse domains, including neural architecture search [32], hyper-parameter tuning [4, 29, 12], and robotic control [18, 5], among others. Nevertheless, in some real-world problems, the stochastic nature of the optimization process, such as machining error during manufacturing, execution noise of control, or variability in contextual factor, inevitably introduces input randomness, rendering the design parameter $x$ to deviate to $x'$ before evaluation. This deviation produces a fluctuation of function value $y$ and eventually leads to a performance instability of the outcome. In general, the input randomness is determined by the application scenario and can be of arbitrary distribution, even quite complex ones. Moreover, in some cases, we cannot observe the exact deviated input $x'$ but a rough estimation for the input uncertainty. This is quite common for robotics and process controls. For example, consider a robot control task shown in Figure 1a, a drone is sent to a target location $x$ to perform a measurement task. However, due to the execution noise caused by the fuzzy control or a sudden wind, the drone ends up at location $x' \sim P(x)$ and gets a noisy measurement $y = f(x') + \zeta$. Instead of observing the exact value of

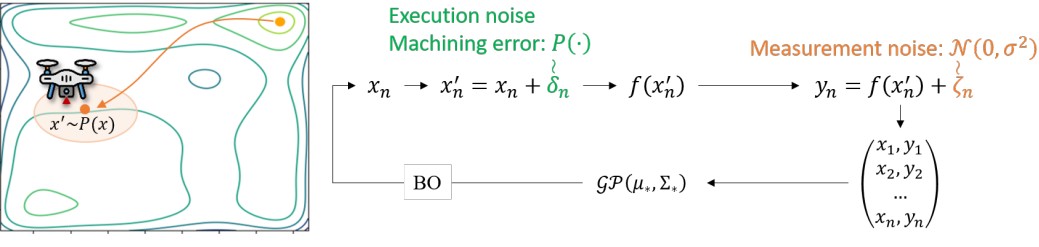

(a) An example case: drone mea­surement with execution noise.

(b) Problem formulation.

Figure 1: Robust Bayesian optimization problem.

$x'$, we only have a coarse estimation of the input uncertainty $P(x)$. The goal is to identify a robust location that gives the maximal expected measurement under the process randomness.

To find a robust optimum, it is crucial to account for input uncertainty during the optimization process. Existing works [24, 3, 7, 10] along this direction assume that the exact input value, *i.e.*, $x'$ in Figure 1b, is observable and construct a surrogate model using these exact inputs. Different techniques are then employed to identify the robust optimum: Nogueira et al. utilize the unscented transform to propagate input uncertainty to the acquisition function [24], while Beland and Nair integrate over the exact GP model to obtain the posterior with input uncertainty [3]. Meanwhile, [7] designs a robust confidence-bounded acquisition and applies min-max optimization to identify the robust optimum. Similarly, [10] constructs an adversarial surrogate with samples from the exact surrogate. These methods work quite well but are constrained by their dependence on observable input values, which may not always be practical.

An alternative approach involves directly modeling the uncertain inputs. A pioneering work by Moreno *et al.* [20] assumes Gaussian input distribution and employs a symmetric Kullback-Leibler divergence (SKL) to measure the distance of input variables. Dallaire *et al.* [13] implement a Gaussian process model with an expected kernel and derive a closed-form solution by restricting the kernel to linear, quadratic, or squared exponential kernels and assuming Gaussian inputs. Nonetheless, the applicability of these methods is limited due to their restrictive Gaussian input distribution assumption and kernel choice. To surmount these limitations, Oliveira *et al.* propose a robust Gaussian process model that incorporates input distribution by computing an integral kernel. Although this kernel can be applied to various distributions and offers a rigorous regret bound, its posterior inference requires a large sampling and can be time-consuming.

In this work, we propose an Arbitrary Input uncertainty Robust Bayesian Optimization algorithm (AIRBO). This algorithm can directly model the uncertain input of arbitrary distribution and propagate the input uncertainty into the surrogate posterior, which can then be used to guide the search for robust optimum. To achieve this, we employ Gaussian Process (GP) as the surrogate and empower its kernel design with the Maximum Mean Discrepancy (MMD), which allows us to comprehensively compare the uncertain inputs in Reproducing Kernel Hilbert Space (RKHS) and accurately quantify the target function under various input uncertainties(Sec. 3.1). Moreover, to stabilize the MMD estimation and accelerate the posterior inference, we utilize Nyström approximation to reduce the space complexity of MMD estimation from $O(m^2)$ to $O(mh)$, where $h \ll m$ (Sec. 3.2). This can substantially improve the parallelization of posterior inference and a rigorous theoretical regret bound is also established under the approximation error (Sec. 4). Comprehensive evaluations on synthetic functions and real problems in Sec.5 demonstrate that our algorithm can efficiently identify robust optimum under complex input uncertainty and achieve state-of-the-art performance.

## 2 Problem Formulation

In this section, we first formulize the robust optimization problem under input uncertainty then briefly review the intuition behind Bayesian Optimization and Gaussian Processes.

### 2.1 Optimization with Input Uncertainty

As illustrated in Figure 1b, we consider an optimization of expensive black-box function: $f(x)$, where $x$ is the *design parameter* to be tuned. At each iteration $n$, we select a new query point $x_n$

according to the optimization heuristics. However, due to the stochastic nature of the process, such as machining error or execution noise, the query point is perturbed to $x'_n$ before the function evaluation. Moreover, we cannot observe the exact value of $x'_n$ and only have a vague probability estimation of its value: $P_{x_n}$. After the function evaluation, we get a noisy measurement $y = f(x'_n) + \zeta_n$, where $\zeta_n$ is homogeneous measurement noise sampled from $\mathcal{N}(0, \sigma^2)$. The goal is to find an optimal design parameter $x^*$ that maximizes the expected function value under input uncertainty:

$$x^* = \arg\max_x \int_{x' \sim P_x} f(x')dx' = \arg\max_x \mathbb{E}_{P_x}[f] \tag{1}$$

Depending on the specific problem and randomness source, the input distribution $P_x$ can be arbitrary in general and even become quite complex sometimes. Here we do not place any additional assumption on them, except assuming we can sample from these input distributions, which can be easily done by approximating it with Bayesian methods and learning a parametric probabilistic model [16]. Additionally, we assume the exact values of $x'$ are inaccessible, which is quite common in some real-world applications, particularly in robotics and process control [25].

## 2.2   Bayesian Optimization

In this paper, we focus on finding the robust optimum with BO. Each iteration of BO involves two key steps: I) fitting a surrogate model and II) maximizing an acquisition function.

**Gaussian Process Surrogate:** To build a sample-efficient surrogate, we choose Gaussian Process (GP) as the surrogate model in this paper. Following [34], GP can be interpreted from a weight-space view: given a set of $n$ observations, $\mathcal{D}_n = \{(x_i, y_i)|i = 1, ..., n\}$. Denote all the inputs as $X \in \mathbb{R}^{D \times n}$ and all the output vector as $y \in \mathbb{R}^{n \times 1}$. We first consider a linear surrogate:

$$f(x) = x^T w, \; y = f(x) + \zeta, \; \zeta \sim \mathcal{N}(0, \sigma^2), \tag{2}$$

where $w$ is the model parameters and $\zeta$ is the observation noise. This model's capacity is limited due to its linear form. To obtain a more powerful surrogate, we can extend it by projecting the input $x$ into a feature space $\phi(x)$. By taking a Bayesian treatment and placing a zero mean Gaussian prior on the weight vector: $w \sim \mathcal{N}(0, \Sigma_p)$, its predictive distribution can be derived as follows (see Section2.1 of [34] for detailed derivation):

$$f_*|x_*, X, y \sim \mathcal{N}\big(\phi^T(x_*)\Sigma_p\phi(X)(A + \sigma_n^2 I)^{-1}y, \\ \phi^T(x_*)\Sigma_p\phi(x_*) - \phi^T(x_*)\Sigma_p\phi(X)(A + \sigma_n^2 I)^{-1}\phi^T(X)\Sigma_p\phi(x_*)\big), \tag{3}$$

where $A = \phi^T(X)\Sigma_p\phi(X)$ and $I$ is a identity matrix. Note the predictive distribution is also a Gaussian and the feature mappings are always in the form of inner product with respect to $\Sigma_p$. This implies we are comparing inputs in a feature space and enables us to apply kernel trick. Therefore, instead of exactly defining a feature mapping $\phi(\cdot)$, we can define a kernel: $k(x, x') = \phi(x)^T\Sigma_p\phi(x') = \psi(x) \cdot \psi(x')$. Substituting it into Eq. 3 gives the vanilla GP posterior:

$$f_*|X, y, X_* \sim \mathcal{N}(\mu_*, \Sigma_*), \text{where } \mu_* = K(X_*, X)[K(X, X) + \sigma_n^2 I]^{-1}y, \\ \Sigma_* = K(X_*, X_*) - K(X_*, X)[K(X, X) + \sigma_n^2 I]^{-1}K(X, X_*). \tag{4}$$

From this interpretation of GP, we note that its core idea is to project the input $x$ to a (possibly infinite) feature space $\psi(x)$ and compare them in the Reproducing Kernel Hilbert Space (RKHS) defined by kernel.

**Acquisition Function Optimization:** Given the posterior of GP surrogate model, the next step is to decide a query point $x_n$. The exploitation and exploration balance is achieved by designing an acquisition function $\alpha(x|\mathcal{D}_n)$. Through numerous acquisition functions exist [28], we follow [25, 7] and adopt the Upper Confidence Bound (UCB) acquisition:

$$\alpha(x|\mathcal{D}_n) = \mu_*(x) + \beta\sigma_*(x), \tag{5}$$

where $\beta$ is a hyper-parameter to control the level of exploration.

## 3   Proposed Method

To cope with randomness during the optimization process, we aim to build a robust surrogate that can directly accept the uncertain inputs of arbitrary distributions and propagate the input uncertainty into

the posterior. Inspired by the weight-space interpretation of GP, we empower GP kernel with MMD to compare the uncertain inputs in RKHS. In this way, the input randomness is considered during the covariance computation and naturally reflected in the resulting posterior , which then can be used to guide the search for a robust optimum(Sec. 3.1). To further accelerate the posterior inference, we employ Nyström approximation to stabilize the MMD estimation and reduce its space complexity (Sec. 3.2).

## 3.1 Modeling the Uncertain Inputs

Assume $P_x \in \mathcal{P}_{\mathcal{X}} \subset \mathcal{P}$ are a set of distribution densities over $\mathbb{R}^d$, representing the distributions of the uncertain inputs. We are interested in building a GP surrogate over the probability space $\mathcal{P}$, which requires to measure the difference between the uncertain inputs.

To do so, we turn to the Integral Probabilistic Metric (IPM) [23]. The basic idea behind IPM is to define a distance measure between two distributions $P$ and $Q$ as the supremum over a class of functions $\mathcal{G}$ of the absolute expectation difference:

$$d(P, Q) = \sup_{g \in \mathcal{G}} |\mathbb{E}_{u \sim P} g(u) - \mathbb{E}_{v \sim Q} g(v)|, \tag{6}$$

where $\mathcal{G}$ is a class of functions that satisfies certain conditions. Different choices of $\mathcal{G}$ lead to various IPMs. For example, if we restrict the function class to be uniformly bounded in RKHS we can get the MMD [15], while a Lipschitz-continuous $\mathcal{G}$ realizes the Wasserstein distance [14].

In this work, we choose MMD as the distance measurement for the uncertain inputs because of its intrinsic connection with distance measurement in RKHS. Given a characteristic kernel $k : \mathbb{R}^d \times \mathbb{R}^d \to \mathbb{R}$ and associate RKHS $\mathcal{H}_k$, define the mean map $\psi : \mathcal{P} \to \mathcal{H}_k$ such that $\langle \psi(P), g \rangle = \mathbb{E}_P[g], \forall g \in \mathcal{H}_k$. The MMD between $P, Q \in \mathcal{P}$ is defined as:

$$\text{MMD}(P, Q) = \sup_{||g||_k \leq 1} [\mathbb{E}_{u \sim P} g(u) - \mathbb{E}_{v \sim Q} g(v)] = ||\psi_P - \psi_Q||, \tag{7}$$

Without any additional assumption on the input distributions, except we can get $m$ samples $\{u_i\}_{i=1}^m, \{v_i\}_{i=1}^m$ from $P, Q$ respectively, MMD can be empirically estimated as follows [21]:

$$\text{MMD}^2(P, Q) \approx \frac{1}{m(m-1)} \sum_{1 \leq i,j \leq m, i \neq j} (k(u_i, u_j) + k(v_i, v_j)) - \frac{2}{m^2} \sum_{1 \leq i,j \leq m} k(u_i, v_j), \tag{8}$$

To integrate MMD into the GP surrogate, we design an MMD-based kernel over $\mathcal{P}$ as follows:

$$\hat{k}(P, Q) = \exp(-\alpha \text{MMD}^2(P, Q)), \tag{9}$$

with a learnable scaling parameter $\alpha$. This is a valid kernel, and universal w.r.t. $C(\mathcal{P})$ under mild conditions (see Theorem 2.2, [11]). Also, it is worth to mention that, to compute the GP posterior, we only need to sample $m$ points from the input distributions, but do not require their corresponding function values.

With the MMD kernel, our surrogate model places a prior $\mathcal{GP}(0, \hat{k}(P_x, P_{x'}))$ and obtain a dataset $\mathcal{D}_n = \{(\hat{x}_i, y_i) | \hat{x}_i \sim P_{x_i}, i = 1, 2, ..., n)\}$. The posterior is Gaussian with mean and variance:

$$\hat{\mu}_n(P_*) = \hat{\mathbf{k}}_n(P_*)^T (\hat{\mathbf{K}}_n + \sigma^2 \mathbf{I})^{-1} \mathbf{y}_n \tag{10}$$

$$\hat{\sigma}_n^2(P_*) = \hat{k}(P_*, P_*) - \hat{\mathbf{k}}_n(P_*)^T (\hat{\mathbf{K}}_n + \sigma^2 \mathbf{I})^{-1} \hat{\mathbf{k}}_n(P_*), \tag{11}$$

where $\mathbf{y}_n := [y_1, \cdots, y_n]^T$, $\hat{\mathbf{k}}_n(P_*) := [\hat{k}(P_*, P_{x_1}), \cdots, \hat{k}(P_*, P_{x_n})]^T$ and $[\hat{\mathbf{K}}_n]_{ij} = \hat{k}(P_{x_i}, P_{x_j})$.

## 3.2 Boosting posterior inference with Nyström Approximation

To derive the posterior distribution of our robust GP surrogate, it requires estimating the MMD between each pair of inputs. Gretton *et al.* prove the empirical estimator in Eq. 8 approximates MMD in a bounded and asymptotic way [15]. However, the sampling size $m$ used for estimation greatly affects the approximation error and insufficient sampling leads to a high estimation variance(ref. Figure 3a).

Such an MMD estimation variance causes numerical instability of the covariance matrix and propagates into the posterior distribution and acquisition function, rendering the search for optimal query point a challenging task. Figure 3b gives an example of MMD-GP posterior with insufficient samples, which produces a noisy acquisition function and impedes the search of optima. Increasing the sampling size can help alleviate this issue. However, the computation and space complexities of the empirical MMD estimator scale quadratically with the sampling size $m$. This leaves us with a dilemma that insufficient sampling results in a highly-varied posterior while a larger sample size can occupy significant GPU memory and reduce the ability for parallel computation.

To reduce the space and computation complexity while retaining a stable MMD estimation, we resort to the Nyström approximation [33]. This method alleviates the computational cost of kernel matrix by randomly selecting $h$ subsamples from the $m$ samples($h \ll m$) and computes an approximated matrix via $\tilde{K} = K_{mh}K_h^+ K_{mh}^T$. Combining this with the MMD definition gives its Nyström estimator:

$$
\begin{aligned}
\text{MMD}^2(P, Q) &= \mathbb{E}_{u,u' \sim P \bigotimes P}[k(u, u')] + \mathbb{E}_{v,v' \sim Q \bigotimes Q}[k(v, v')] - 2\mathbb{E}_{u,v \sim P \bigotimes Q}[k(u, v)] \\
&\approx \frac{1}{m^2}\mathbf{1}_m^T U \mathbf{1}_m + \frac{1}{m^2}\mathbf{1}_m^T V \mathbf{1}_m - \frac{2}{m^2}\mathbf{1}_m^T W \mathbf{1}_m \\
&\approx \frac{1}{m^2}\mathbf{1}_m^T U_{mh}U_h^+ U_{mh}^T \mathbf{1}_n + \frac{1}{m^2}\mathbf{1}_m^T V_{mh}V_h^+ V_{mh}^T \mathbf{1}_m - \frac{2}{m^2}\mathbf{1}_m^T W_{mh}W_h^+ W_{mh}^T \mathbf{1}_m
\end{aligned}
\tag{12}
$$

where $U = K(\mathbf{u}, \mathbf{u}')$, $V = K(\mathbf{v}, \mathbf{v}')$, $W = K(\mathbf{u}, \mathbf{v})$ are the kernel matrices, $\mathbf{1}_m$ represents a m-by-1 vector of ones, $m$ defines the sampling size and $h$ controls the sub-sampling size. Note that this Nyström estimator reduces the space complexity of posterior inference from $O(MNm^2)$ to $O(MNmh)$, where $M$ and $N$ are the numbers of training and testing samples, $m$ is the sampling size for MMD estimation while $h \ll m$ is the sub-sampling size. This can significantly boost the posterior inference of robust GP by allowing more inference to run in parallel on GPU.

# 4 Theoretical Analysis

Assume $x \in \mathcal{X} \subset \mathbb{R}^d$, and $P_x \in \mathcal{P}_{\mathcal{X}} \subset \mathcal{P}$ are a set of distribution densities over $\mathbb{R}^d$, representing the distribution of the noisy input. Given a characteristic kernel $k : \mathbb{R}^d \times \mathbb{R}^d \to \mathbb{R}$ and associate RKHS $\mathcal{H}_k$, we define the mean map $\psi : \mathcal{P} \to \mathcal{H}_k$ such that $\langle \psi(P), g \rangle = \mathbb{E}_P[g], \forall g \in \mathcal{H}_k$.

We consider a more general case. Choosing any suitable functional $L$ such that $\hat{k}(P, P') := L(\psi_P, \psi_{P'})$ is a positive-definite kernel over $\mathcal{P}$, for example the linear kernel $\langle \psi_P, \psi_{P'} \rangle_k$ and radial kernels $\exp(-\alpha\|\psi_P - \psi_{P'}\|_k^2)$ using the MMD distance as a metric. Such a kernel $\hat{k}$ is associated with a RKHS $\mathcal{H}_{\hat{k}}$ containing functions over the space of probability measures $\mathcal{P}$.

One important theoretical guarantee to conduct $\mathcal{GP}$ model is that our object function can be approximated by functions in $\mathcal{H}_{\hat{k}}$, which relies on the universality of $\hat{k}$. Let $C(\mathcal{P})$ be the class of continuous functions over $\mathcal{P}$ endowed with the topology of weak convergence and the associated Borel $\sigma$-algebra, and we define $\hat{f} \in C(\mathcal{P})$ such that

$$\hat{f}(P) := \mathbb{E}_P[f], \forall P \in \mathcal{P},$$

which is just our object function, For $\hat{k}$ be radial kernels, it has been shown that $\hat{k}$ is universal w.r.t $C(\mathcal{P})$ given that $\mathcal{X}$ is compact and the mean map $\psi$ is injective [11, 22]. For $\hat{k}$ be linear kernel which is not universal, it has been shown in Lemma 1, [26] that $\hat{f} \in \mathcal{H}_{\hat{k}}$ if and only if $f \in \mathcal{H}$ and further $\|\hat{f}\|_{\hat{k}} = \|f\|_k$. Thus, in the remain of this chapter, we may simply assume $\hat{f} \in \mathcal{H}_{\hat{k}}$.

Suppose we have an approximation kernel function $\tilde{k}(P, Q)$ near to the exact kernel function $\hat{k}(P, Q)$. The mean $\hat{\mu}_n(p_*)$ and variance $\hat{\sigma}_n^2(p_*)$ are approximated by

$$\tilde{\mu}_n(P_*) = \tilde{\mathbf{k}}_n(P_*)^T(\tilde{\mathbf{K}}_n + \sigma^2\mathbf{I})^{-1}\mathbf{y}_n \tag{13}$$

$$\tilde{\sigma}_n^2(P_*) = \tilde{k}(P_*, P_*) - \tilde{\mathbf{k}}_n(P_*)^T(\tilde{\mathbf{K}}_n + \sigma^2\mathbf{I})^{-1}\tilde{\mathbf{k}}_n(P_*), \tag{14}$$

where $\mathbf{y}_n := [y_1, \cdots, y_n]^T$, $\tilde{\mathbf{k}}_n(P_*) := [\tilde{k}(P_*, P_1), \cdots, \tilde{k}(P_*, P_n)]^T$ and $[\tilde{\mathbf{K}}_n]_{ij} = \tilde{k}(P_i, P_j)$.

The maximum information gain corresponding to the kernel $\hat{k}$ is denoted as

$$\hat{\gamma}_n := \sup_{\mathcal{R} \in \mathcal{P}_{\mathcal{X}}; |\mathcal{R}|=n} \hat{I}(\mathbf{y}_n; \hat{\mathbf{f}}_n | \mathcal{R}) = \frac{1}{2} \ln \det(\mathbf{I} + \sigma^{-2}\hat{\mathbf{K}}_n),$$

Denote $e(P,Q) = \hat{k}(P,Q) - \tilde{k}(P,Q)$ as the error function when estimating the kernel $\hat{k}$. We suppose $e(P,Q)$ has an upper bound with high probability:

**Assumption 1.** *For any $\varepsilon > 0$, $P, Q \in \mathcal{P}_{\mathcal{X}}$, we may choose an estimated $\tilde{k}(P,Q)$ such that the error function $e(P,Q)$ can be upper-bounded by $e_\varepsilon$ with probability at least $1 - \varepsilon$, that is, $\mathbb{P}\left(|e(P,Q)| \leq e_\varepsilon\right) > 1 - \varepsilon$.*

**Remark.** Note that this assumption is standard in our case: we may assume $\max_{x \in \mathcal{X}} \|\phi\|_k \leq \Phi$, where $\phi$ is the feature map corresponding to the $k$. Then when using empirical estimator, the error between $\mathrm{MMD}_{\mathrm{empirical}}$ and MMD is controlled by $4\Phi\sqrt{2\log(6/\varepsilon)m^{-1}}$ with probability at least $1 - \varepsilon$ according to Lemma E.1, [8]. When using the Nyström estimator, the error has a similar form as the empirical one, and under mild conditions, when $h = O(\sqrt{m}\log(m))$, we get the error of the order $O(m^{-1/2}\log(1/\varepsilon))$ with probability at least $1 - \varepsilon$. One can check more details in Lemma 1.

Now we restrict our Gaussian process in the subspace $\mathcal{P}_{\mathcal{X}} = \{P_x, x \in \mathcal{X}\} \subset \mathcal{P}$. We assume the observation $y_i = f(x_i) + \zeta_i$ with the noise $\zeta_i$. The input-induced noise is defined as $\Delta f_{p_{x_i}} := f(x_i) - \mathbb{E}_{P_{x_i}}[f] = f(x_i) - \hat{f}(P_{x_i})$. Then the total noise is $y_i - \mathbb{E}_{P_{x_i}}[f] = \zeta_i + \Delta f_{p_{x_i}}$. We can state our main result, which gives a cumulative regret bound under inexact kernel calculations,

**Theorem 1.** *Let $\delta > 0$, $f \in \mathcal{H}_k$, and the corresponding $\|\hat{f}\|_{\hat{k}} \leq b, \max_{x \in \mathcal{X}} |f(x)| = M$. Suppose the observation noise $\zeta_i = y_i - f(x_i)$ is $\sigma_\zeta$-sub-Gaussian, and thus with high probability $|\zeta_i| < A$ for some $A > 0$. Assume that both $k$ and $P_x$ satisfy the conditions for $\Delta f_{P_x}$ to be $\sigma_E$-sub-Gaussian, for a given $\sigma_E > 0$. Then, under Assumption 1 with $\varepsilon > 0$ and corresponding $e_\varepsilon$, setting $\sigma^2 = 1 + \frac{2}{n}$, running Gaussian Process with acquisition function*

$$\tilde{\alpha}(x|\mathcal{D}_n) = \tilde{\mu}_n(P_x) + \beta_n \tilde{\sigma}_n(P_x) \tag{15}$$

$$\textit{where } \beta_n = \left(b + \sqrt{\sigma_E^2 + \sigma_\zeta^2}\sqrt{2\left(\hat{\gamma}_n + 1 - \ln\delta\right)}\right),$$

*we have that the uncertain-inputs cumulative regret satisfies:*

$$\tilde{R}_n \in O\left(\sqrt{n\hat{\gamma}_n(\hat{\gamma}_n - \ln\delta)} + n^2\sqrt{(\hat{\gamma}_n - \ln\delta)e_\varepsilon} + n^3 e_\varepsilon\right) \tag{16}$$

*with probability at least $1 - \delta - n\varepsilon$. Here $\tilde{R}_n = \sum_{t=1}^n \tilde{r}_t$, and $\tilde{r}_t = \max_{x \in \mathcal{X}} \mathbb{E}_{P_x}[f] - \mathbb{E}_{P_{x_t}}[f]$*

The proof of our main theorem 1 can be found in appendix B.3.

The assumption that $\zeta_i$ is $\sigma_\zeta$-sub-Gaussian is standard in $\mathcal{GP}$ fields. The assumption that $\Delta f_{P_x}$ is $\sigma_E$-sub-Gaussian can be met when $P_x$ is uniformly bounded or Gaussian, as stated in Proposition 3, [26]. Readers may check the definition of sub-Gaussian in appendix, Definition 1.

To achieve an regret of order $\tilde{R}_n \in O(\sqrt{n}\hat{\gamma}_n)$, the same order as the exact Improved $\mathcal{GP}$ regret (23), and ensure this with high probability, we need to take $\varepsilon = O(\delta/n)$, $e_\varepsilon = O(n^{-\frac{5}{2}}\hat{\gamma}_n(\hat{\gamma}_n^{-2} \wedge n^{-\frac{1}{2}}))$, and this requires a sample size $m$ of order $O(n^5\hat{\gamma}_n^{-2}(\hat{\gamma}_n^4 \vee n)\log(n))$ for MCMC approximation, or with a same sample size $m$ and a subsample size $h$ of order $O(n^{\frac{5}{2}+\nu}\hat{\gamma}_n^{-1-\nu}(\hat{\gamma}_n^2 \vee n^{\frac{1}{2}}))$ for Nyström approximation with some $\nu > 0$. Note that (16) only offers an upper bound for cumulative regret, in real applications the calculated regret may be much smaller than this bound, as the approximation error $e_\epsilon$ can be fairly small even with a few samples when the input noise is relatively weak.

To analysis the exact order of $\hat{\gamma}_n$ could be difficult, as it is influenced by the specific choice of embedding kernel $k$ and input uncertainty distributions $P_{x_i}, x_i \in \mathcal{X}$. Nevertheless, we can deduce the following result for a wide range of cases, showing that cumulative regret is sub-linear under mild conditions. One can check the proof in appendix B.4.

**Theorem 2** (Bounding the Maximum information gain). *Suppose $k$ is $r$-th differentiable with bounded derivatives and translation invariant, i.e., $k(x,y) = k(x-y, 0)$. Suppose the input uncertainty is i.i.d., that is, the noised input density satisfies $P_{x_i}(x) = P_0(x - x_i), \forall x_i \in \mathcal{X}$. Then if the space $\mathcal{X}$ is compact in $\mathbb{R}^d$, the maximum information gain $\hat{\gamma}_n$ satisfies*

$$\hat{\gamma}_n = O(n^{\frac{d(d+1)}{r+d(d+1)}}\log(n)).$$

*Thus, when $r > d(d+1)$, the accumulate regret is sub-linear respect to $n$, with sufficiently small $e_\varepsilon$.*

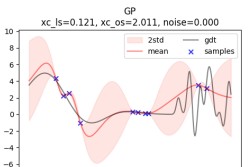 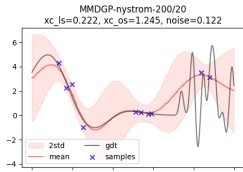 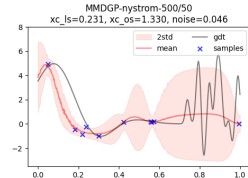 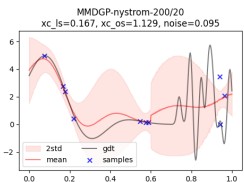

(a) GP posterior with a Gaussian input uncertainty $P = \mathcal{N}(0, 0.01)$.

(b) MMDGP posterior with an input uncertainty $P = \mathcal{N}(0, 0.01)$.

(c) MMDGP posterior with a variance-changing beta distribution.

(d) MMDGP posterior with a Chi-squared distribution of changing DoF.

Figure 2: Modeling results under different types of input uncertainties.

## 5 Evaluation

In this section, we first experimentally demonstrate AIRBO's ability to model uncertain inputs of arbitrary distributions, then validate the Nyström-based inference acceleration for GP posterior, followed by experiments on robust optimization of synthetic functions and real-world benchmark.

### 5.1 Robust Surrogate

**Modeling arbitrary uncertain inputs**: We demonstrate MMDGP's capabilities by employing an RKHS function as the black-box function and randomly selecting 10 samples from its input domain. Various types of input randomness are introduced into the observation and produce training datasets of $\mathcal{D} = \{(x_i, f(x_i + \delta_i)) | \delta_i \sim P_{x_i}\}_{i=1}^{10}$ with different $P_x$ configurations. Figure 2a and 2b compare the modeling results of a conventional GP and MMDGP under a Gaussian input uncertainty $P_x = \mathcal{N}(0, 0.01^2)$. We observe that the GP model appears to overfit the observed samples without recognizing the input uncertainty, whereas MMDGP properly incorporates the input randomness into its posterior.

To further examine our model's ability under complex input uncertainty, we design the input distribution to follow a beta distribution with input-dependent variance: $P_x = beta(\alpha = 0.5, \beta = 0.5, \sigma = 0.9(\sin 4\pi x + 1))$. The MMDGP posterior is shown in Figure 2c. As the input variance $\sigma$ changes along $x$, inputs from the left and right around a given location $x_i$ yield different MMD distances, resulting in an asymmetric posterior (*e.g.*, around $x = 0.05$ and $x = 0.42$). This suggests that MMDGP can precisely model the multimodality and asymmetry of the input uncertainty.

Moreover, we evaluated MMDGP using a step-changing Chi-squared distribution $P_x = \chi^2(g(x), \sigma = 0.01)$, where $g(x) = 0.5$ if $x \in [0, 0.6]$, and $g(x) = 7.0$ otherwise. This abrupt change in $g(x)$ significantly alters the input distribution from a sharply peaked distribution to a flat one with a long tail. Figure 2d illustrates that our model can accurately capture this distribution shape variation, as evidenced by the sudden posterior change around $x = 0.6$. This demonstrates our model can thoroughly quantify the characteristics of complex input uncertainties.

**Comparing with the other surrogate models:** We also compare our model with the other surrogate models under the step-changing Chi-squared input distribution. The results are reported in Figure 7 and they demonstrate our model outperforms obviously under such a complex input uncertainty (see Appendix D.1 for more details)

### 5.2 Accelerating the Posterior Inference

**Estimation variance of MMD:** We first examine the variance of MMD estimation by employing two beta distributions $P = beta(\alpha = 0.4, \beta = 0.2, \sigma = 0.1)$ and $Q = beta(\alpha = 0.4, \beta = 0.2, \sigma = 0.1) + c$, where $c$ is an offset value. Figure 3a shows the empirical MMDs computed via Eq. 8 with varying sampling sizes as $Q$ moves away from $P$. We find that a sampling size of 20 is inadequate, leading to high estimation variance, and increasing the sampling size to 100 stabilizes the estimation.

We further utilize this beta distribution $P$ as the input distribution and derive the MMDGP posterior via empirical estimator in Figure 3b. Note that the MMD instability caused by inadequate sampling subsequently engenders a fluctuating posterior and culminates in a noisy acquisition function, which prevents the acquisition optimizer (*e.g.*, L-BFGS-B in this experiment) from identifying the optima. Although Figure 3c shows that this issue can be mitigated by using more samples during empirical

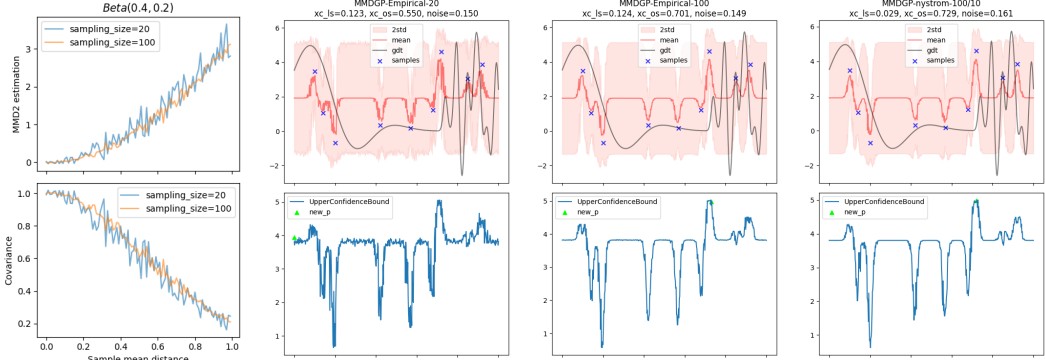

(a) The empirical MMD and covariance values between two beta distributions $P$ and $Q$ as $Q$ moves away from $P$.

(b) The noisy posterior derived from a sampling size of 20 (upper) traps the acq. optimizer at $x = 0$ (lower).

(c) The posterior becomes smoother with a sampling size of 100 and acq. optimizer can easily identify the optima.

(d) The Nyström estimator with less memory consumption also produces a smooth posterior that is easy to optimize.

Figure 3: The posterior derived from the empirical and Nyström MMD approximators with varying sampling sizes.

Table 1: Performance of Posterior inference for 512 samples.

| Method | Sampling Size | Sub-sampling Size | Inference Time (seconds) | Batch Size (samples) |
|---|---|---|---|---|
| Empirical | 20 | - | $1.143 \pm 0.083$ | 512 |
| Empirical | 100 | - | $8.117 \pm 0.040$ | 128 |
| Empirical | 1000 | - | $840.715 \pm 2.182$ | 1 |
| Nystrom | 100 | 10 | $0.780 \pm 0.001$ | 512 |
| Nystrom | 1000 | 100 | $21.473 \pm 0.984$ | 128 |

MMD estimation, it is crucial to note that a larger sampling size significantly increases GPU memory usage because of its quadratic space complexity of $O(MNm^2)$ ($M$ and $N$ are the sample number of training and testing, $m$ is the sampling size for MMD estimation). This limitation severely hinders parallel inference for multiple samples and slows the overall speed of posterior computation.

Table 1 summarizes the **inference time** of MMDGP posteriors at 512 samples with different sampling sizes. We find that, for beta distribution defined in this experiment, the Nyström MMD estimator with a sampling size of 100 and sub-sampling size of 10 already delivers a comparable result to the empirical estimator with 100 samples (as seen in the acquisition plot of Figure 3d). Also, the inference time is reduced from 8.117 to 0.78 seconds by enabling parallel computation for more samples. For the cases that require much more samples for MMD estimation (*e.g.*, the input distribution is quite complex or high-dimensional), this Nyström-based acceleration can have a more pronounced impact.

**Effect of Nyström estimator on optimization:** To investigate the effect of Nyström estimator on optimization, we also perform an ablation study in Appendix D.2, the results in Figure 8 suggest that Nyström estimator slightly degrades the optimization performance but greatly improves the inference efficiency.

## 5.3 Robust Optimization

**Experiment setup:** To experimentally validate AIRBO's performance, we implement our algorithm [1] based on BoTorch [2] and employ a linear combination of multiple rational quadratic kernels [6] to compute the MMD as Eq. 9. We compare our algorithm with several baselines: 1) **uGP-UCB** [26] is a closely related work that employs an integral kernel to model the various input distributions. It has a quadratic inference complexity of $O(MNm^2)$, where $M$ and $N$ are the sample numbers of the training and testing set, and $m$ indicates the sampling size of the integral kernel. 3)**GP-UCB** is the standard GP with UCB acquisition, which represents a broad range of existing methods that

---

[1]The code will be available on `https://github.com/huawei-noah/HEBO`, and more implementation details can be found in Appendix C.1.

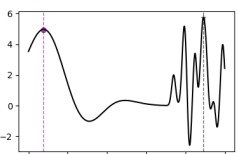 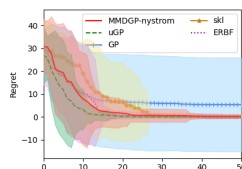 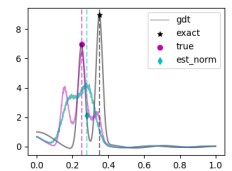 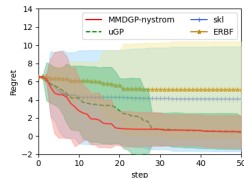

(a) The RKHS function    (b) Robust regrets of RKHS function.    (c) The double-peak function    (d) Robust regrets of double-peak function

Figure 4: Robust optimization results on synthetic functions.

focus on non-robust optimization. 3) **SKL-UCB** employs symmetric Kullback-Leibler divergence to measure the distance between the uncertain inputs [20]. Its closed-form solution only exists if the input distributions are the Gaussians. 4) **ERBF-UCB** is the robust GP with the expected Radial Basis Function kernel proposed in [13]. It computes the expected kernel under input distribution using the Gaussian integrals. Assuming the input distributions are sub-Gaussians, this method can efficiently find the robust optimum. Since all the methods use UCB acquisition, we simply distinguish them by their surrogate names in the following tests.

At the end of the optimization, each algorithm needs to decide a final *outcome* $x_n^r$, perceived to be the robust optimum under input uncertainty at step $n$. For a fair comparison, we employ the same outcome policy across all the algorithms: $x_n^r = \arg\max_{x \in \mathcal{D}_n} \hat{\mu}_*(x)$, where $\hat{\mu}_*(x)$ is the posterior mean of robust surrogate at $x$ and $\mathcal{D}_n = \{(x_i, f(x_i + \delta_i)) | \delta_i \sim P_{x_i}\}$ are the observations so far. The optimization performance is measured in terms of **robust regret** as follows:

$$r(x_n^r) = \mathbb{E}_{\delta \sim P_{x^*}}[f(x^* + \delta)] - \mathbb{E}_{\delta \sim P_{x_n^r}}[f(x_n^r + \delta)], \tag{17}$$

where $x^*$ is the global robust optimum and $x_n^r$ represents the outcome point at step $n$. For each algorithm, we repeat the optimization process 12 times and compare the average robust regret.

**1D RKHS function:** We begin the optimization evaluation with an RKHS function that is widely used in previous BO works [1, 24, 10]. Figure 4a shows its exact global optimum resides at $x = 0.892$ while the robust optimum is around $x = 0.08$ when the inputs follow a Gaussian distribution $\mathcal{N}(0, 0.01^2)$. According to Figure 4b, all the robust BO methods work well with Gaussian uncertain inputs and efficiently identify the robust optimum, but the GP-UCB stagnates at a local optimum due to its neglect of input uncertainty. Also, we notice the regret of our method decrease slightly slower than uGP works in this low-dimensional and Gaussian-input case, but later cases with higher dimension and more complex distribution show our method is more stable and efficient.

**1D double-peak function:** To test with more complex input uncertainty, we design a blackbox function with double peaks and set the input distribution to be a multi-modal distribution $P_x = beta(\alpha = 0.4, \beta = 0.2, \sigma = 0.1)$. Figure 4c shows the blackbox function (black solid line) and the corresponding function expectations estimated numerically via sampling from the input distribution (*i.e.*, the colored lines). Note the true robust optimum is around $x = 0.251$ under the beta distribution, but an erroneous location at $x = 0.352$ may be determined if the input uncertainty is incorrectly presumed to be Gaussian. This explains the results in Figure 4d: the performance of SKL-UCB and ERBF-UCB are sub-optimal due to their misidentification of inputs as Gaussian variables, while our method accurately quantifies the input uncertainty and outperforms the others.

**10D bumped-bowl function:** we also extend our evaluation to a 10D bumped-bowl function [27] under a concatenated circular distribution. Figure 9 demonstrates AIRBO scales efficiently to high dimension and outperforms the others under complex input uncertainty(see Appendix D.3).

**Robust robot pushing:** To evaluate AIRBO in a real-world problem, we employ a robust robot pushing benchmark from [31], in which a ball is placed at the origin point of a 2D space and a robot learns to push it to a predefined target location $(g_x, g_y)$. This benchmark takes a 3-dimensional input $(r_x, r_y, r_t)$, where $r_x, r_y \in [-5, +5]$ are the 2D coordinates of the initial robot location and $r_t \in [0, 30]$ controls the push duration. We set four targets in separate quadrants, *i.e.*, $g1 = (-3, -3), g_2 = (-3, 3), g_3 = (4.3, 4.3)$, and a "twin" target at $g_3' = (5.1, 3.0)$, and describe the input uncertainty via a two-component Gaussian Mixture Model (defined in Appendix D.4). Following [7, 10], this blackbox benchmark outputs the minimum distance to these 4 targets under squared and linear distances: $loss = \min(d^2(g_1, l), d(g_2, l), d(g_3, l), d(g_3', l))$, where $d(g_i, l)$ is the Euclidean distance between the ball's ending location $l$ and the $i$-th target. This produces a loss landscape as

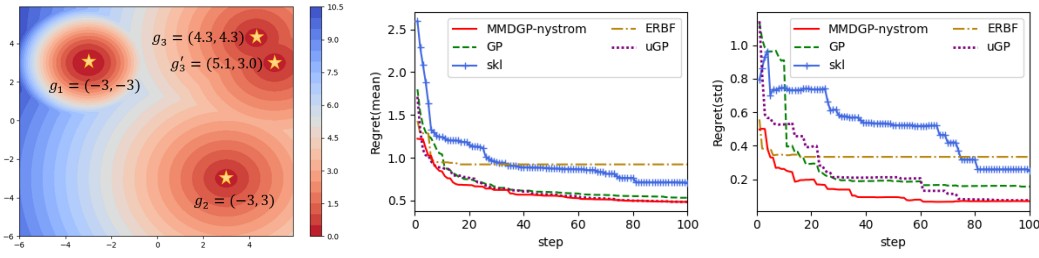

(a) Contour of the robot push world

(b) Robust regrets of different algorithms

Figure 5: Robust optimization of the robot push problem.

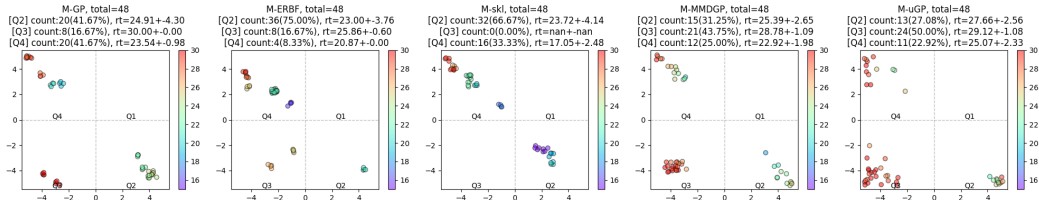

Figure 6: The robot's initial locations and push times found by different algorithms

shown in Figure 5a. Note that $g_2$ is a more robust target than $g_1$ because of its linear-form distance while pushing the ball to quadrant I is the best choice as the targets, $g_3$ and $g_3'$, match the dual-mode pattern of the input uncertainty. According to Figure 5b, our method obviously outperforms the others because it efficiently quantifies the multimodal input uncertainty. This can be further evidenced by the push configurations found by different algorithms in Figure 6, in which each dot represents the robot's initial location and its color represents the push duration. We find that AIRBO successfully recognizes the targets in quadrant I as an optimal choice and frequently pushes from quadrant III to quadrant I. Moreover, the pushes started close to the origin can easily go far away under input variation, so our method learns to push the ball from a corner with a long push duration, which is more robust in this case.

## 6 Discussion and Conclusion

In this work, we generalize robust Bayesian Optimization to an uncertain input setting. The weight-space interpretation of GP inspires us to empower the GP kernel with MMD and build a robust surrogate for uncertain inputs of arbitrary distributions. We also employ the Nyström approximation to boost the posterior inference and provide theoretical regret bound under approximation error. The experiments on synthetic blackbox function and benchmarks demonstrate our method can handle various input uncertainty and achieve state-of-the-art optimization performance.

There are several interesting directions that worth to explore: though we come to current MMD-based kernel from the weight-space interpretation of GP and the RKHS realization of MMD, our kernel design exhibits a deep connection with existing works on kernel over probability measures [22, 11]. Along this direction, as our theoretic regret analysis in Section 4 does not assume any particular form of kernel and the Nyström acceleration can also be extended to the other kernel computation, it is possible that AIRBO can be further generalized to a more rich family of kernels. Moreover, the MMD used in our kernel is by no means limited to its RKHS realization. In fact, any function class $\mathcal{F}$ that comes with uniform convergence guarantees and is sufficiently rich can be used, which renders different realizations of MMD. With proper choice of function class $\mathcal{F}$, MMD can be expressed as the Kolmogorov metric or other Earth-mover distances [15]. It is also interesting to extend AIRBO with the other IPMs.

## 7 Acknowledgements

We sincerely thank Yanbin Zhu and Ke Ma for their help on formulating the problem. Also, a heartfelt appreciation goes to Lu Kang for her constant encouragement and support throughout this work.

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

## A  Nyström Estimator Error Bound

Nyström estimator can easily approximate the kernel mean embedding $\psi_{p_1}, \psi_{p_2}$ as well as the MMD distance between two distribution density $p_1$ and $p_2$. We need first assume the boundedness of the feature map to the kernel $k$:

**Assumption 2.** *There exists a positive constant $K \leq \infty$ such that $\sup_{x \in \mathcal{X}} \|\phi(x)\| \leq K$*

The true MMD distance between $p_1$ and $p_2$ is denoted as $\mathrm{MMD}(p_1, p_2)$. The estimated MMD distance when using a Nyström sample size $m_i$, sub-sample size $h_i$ for $p_i$ respectively, is denoted as $\mathrm{MMD}_{(p_i, m_i, h_i)}$. Then the error

$$\mathrm{Err}_{(p_i, m_i, h_i)} := |\mathrm{MMD}(p_1, p_2) - \mathrm{MMD}_{(p_i, m_i, h_i)}|$$

and now we have the lemma from Theorem 5.1 in [8]

**Lemma 1.** *Let Assumption 2 hold. Furthermore, assume that for $i \in 1, 2$, the data points $X_1^i, \cdots, X_{m_i}^i$ are drawn i.i.d. from the distribution $\rho_i$ and that $h_i \leq m_i$ sub-samples $\tilde{X}_1^i, \cdots, \tilde{X}_{h_i}^i$ are drawn uniformly with replacement from the dataset $\{X_1^i, \cdots, X_{m_i}^i\}$. Then, for any $\delta \in (0, 1)$, it holds with probability at least $1 - 2\delta$*

$$\mathrm{Err}_{(p_i, m_i, h_i)} \leq \sum_{i=1,2} \left( \frac{c_1}{\sqrt{m_i}} + \frac{c_2}{h_i} + \frac{\sqrt{\log(h_i/\delta)}}{h_i} \sqrt{\mathcal{N}^{p_i}\left(\frac{12K^2 \log(h_i/\delta)}{h_i}\right)} \right),$$

*provided that, for $i \in \{1, 2\}$,*

$$h_i \geq \max(67, 12K^2 \|C_i\|_{\mathcal{L}(\mathcal{H})}^{-1}) \log(h_i/\delta)$$

*where $c_1 = 2K\sqrt{2\log(6/\delta)}, c_2 = 4\sqrt{3}K\log(12/\delta)$ and $c_4 = 6K\sqrt{\log(12/\delta)}$. The notation $\mathcal{N}^{p_i}$ denotes the effective dimension associated to the distribution $p_k$.*

*Specifically, when the effective dimension $\mathcal{N}$ satisfies, for some $c \geq 0$,*

- *either $\mathcal{N}^{\rho_i}(\sigma^2) \leq c\sigma^{2-\gamma}$ for some $\gamma \in (0, 1)$,*

- *or $\mathcal{N}^{\rho_i}(\sigma^2) \leq \log(1 + c/\sigma^2)/\beta$, for some $\beta > 0$.*

*Then, choosing the subsample size $m$ to be*

- *$h_i = m_i^{1/(2-\gamma)} \log(m_i/\delta)$ in the first case*

- *or $h_i = \sqrt{m_i} \log(\sqrt{m_i} \max(1/\delta, c/(6K^2)))$ in the second case,*

*we get $\mathrm{Err}_{(\rho_i, m_i, h_i)} = O(1/\sqrt{m_i})$*

## B  Proofs of Section 4

### B.1  Exact Kernel Uncertainty $\mathcal{GP}$ Formulating

Following the same notation in Section 4, now we can construct a Gaussian process $\mathcal{GP}(0, \hat{k})$ modelling functions over $\mathcal{P}$. This $\mathcal{GP}$ model can then be applied to learn $\hat{f}$ from a given set of observations $\mathcal{D}_n = \{(P_i, y_i)\}_{i=1}^n$. Under zero mean condition, the value of $\hat{f}(P_*)$ for a given $P_* \in \mathcal{P}$ follows a Gaussian posterior distribution with

$$\hat{\mu}_n(P_*) = \hat{\mathbf{k}}_n(P_*)^T (\hat{\mathbf{K}}_n + \sigma^2 \mathbf{I})^{-1} \mathbf{y}_n \tag{18}$$

$$\hat{\sigma}_n^2(P_*) = \hat{k}(P_*, P_*) - \hat{\mathbf{k}}_n(P_*)^T (\hat{\mathbf{K}}_n + \sigma^2 \mathbf{I})^{-1} \hat{\mathbf{k}}_n(P_*), \tag{19}$$

where $\mathbf{y}_n := [y_1, \cdots, y_n]^T$, $\hat{\mathbf{k}}_n(P_*) := [\hat{k}(P_*, P_1), \cdots, \hat{k}(P_*, P_n)]^T$ and $[\hat{\mathbf{K}}_n]_{ij} = \hat{k}(P_i, P_j)$.

Now we restrict our Gaussian process in the subspace $\mathcal{P}_{\mathcal{X}} = \{P_x, x \in \mathcal{X}\} \subset \mathcal{P}$. We assume the observation $y_i = f(x_i) + \zeta_i$ with the noise $\zeta_i$. The input-induced noise is defined as $\Delta f_{p_{x_i}} := f(x_i) - \mathbb{E}_{P_{x_i}}[f] = f(x_i) - \hat{f}(P_{x_i})$. Then the total noise is $y_i - \mathbb{E}_{P_{x_i}}[f] = \zeta_i + \Delta f_{p_{x_i}}$. To formulate the regret bounds, we introduce the information gain and estimated information gain given any $\{P_t\}_{t=1}^n \subset \mathcal{P}$:

$$\hat{I}(\mathbf{y}_n; \hat{\mathbf{f}}_n | \{P_t\}_{t=1}^n) := \frac{1}{2} \ln \det(\mathbf{I} + \sigma^{-2} \hat{\mathbf{K}}_n), \tag{20}$$

$$\tilde{I}(\mathbf{y}_n; \hat{\mathbf{f}}_n | \{P_t\}_{t=1}^n) := \frac{1}{2} \ln \det(\mathbf{I} + \sigma^{-2} \tilde{\mathbf{K}}_n), \tag{21}$$

and the maximum information gain is defined as $\hat{\gamma}_n := \sup_{\mathcal{R} \in \mathcal{P}_{\mathcal{X}}; |\mathcal{R}| = n} \hat{I}(\mathbf{y}_n; \hat{\mathbf{f}}_n | \mathcal{R})$. Here $\hat{\mathbf{f}}_n := [\hat{f}(p_1), \cdots, \hat{f}(p_n)]^T$.

We define the sub-Gaussian condition as follows:

**Definition 1.** *For a given $\sigma_\xi > 0$, a real-valued random variable $\xi$ is said to be $\sigma_\xi$-sub-Gaussian if:*

$$\forall \lambda \in \mathbb{R}, \mathbb{E}[e^{\lambda \xi}] \leq e^{\lambda^2 \sigma_\xi^2 / 2}$$

Now we can state the lemma for bounding the uncertain-inputs regret of exact kernel evaluations, which is originally stated in Theorem 5 in [26].

**Lemma 2.** *Let $\delta \in (0, 1)$, $f \in \mathcal{H}_k$, and the corresponding $\|\hat{f}\|_{\hat{k}} \leq b$. Suppose the observation noise $\zeta_i = y_i - f(x_i)$ is conditionally $\sigma_\zeta$-sub-Gaussian. Assume that both $k$ and $P_x$ satisfy the conditions for $\Delta f_{P_x}$ to be $\sigma_E$-sub-Gaussian, for a given $\sigma_E > 0$. Then, we have the following results:*

- *The following holds for all $x \in \mathcal{X}$ and $t \geq 1$:*

$$|\hat{\mu}_n(P_x) - \hat{f}(P_x)| \leq \left( b + \sqrt{\sigma_E^2 + \sigma_\zeta^2} \sqrt{2 \left( \hat{I}(\mathbf{y}_n; \hat{\mathbf{f}}_n | \{P_t\}_{t=1}^n) + 1 + \ln(1/\delta) \right)} \right) \hat{\sigma}_n(P_x) \tag{22}$$

- *Running with upper confidence bound (UCB) acquisition function $\alpha(x|\mathcal{D}_n) = \hat{\mu}_n(P_x) + \hat{\beta}_n \hat{\sigma}_n(P_x)$ where*

$$\hat{\beta}_n = b + \sqrt{\sigma_E^2 + \sigma_\zeta^2} \sqrt{2 \left( \hat{I}(\mathbf{y}_n; \hat{\mathbf{f}}_n | \{P_t\}_{t=1}^n) + 1 + \ln(1/\delta) \right)},$$

*and set $\sigma^2 = 1 + 2/n$, the uncertain-inputs cumulative regret satisfies:*

$$\hat{R}_n \in O(\sqrt{n \hat{\gamma}_n}(b + \sqrt{\hat{\gamma}_n + \ln(1/\delta)})) \tag{23}$$

*with probability at least $1 - \delta$.*

Note that although the original theorem restricted to the case when $\hat{k}(p, q) = \langle \psi_P, \psi_Q \rangle_k$, the results can be easily generated to other kernels over $\mathcal{P}$, as long as its universal w.r.t $C(\mathcal{P})$ given that $\mathcal{X}$ is compact and the mean map $\psi$ is injective [11, 22].

## B.2 Error Estimates for Inexact Kernel Approximation

Now let us derivative the inference under the introduce of inexact kernel estimations.

**Theorem 3.** *Under the Assumption 1 for $\varepsilon > 0$, let $\tilde{\mu}_n, \tilde{\sigma}_n, \tilde{I}(\mathbf{y}_n; \hat{\mathbf{f}}_n | \{P_t\}_{t=1}^n)$ as defined in (13),(14),(21) respectively, and $\hat{\mu}_n, \hat{\sigma}_n, \hat{I}(\mathbf{y}_n; \hat{\mathbf{f}}_n | \{P_t\}_{t=1}^n)$ as defined in (18),(19),(20). Assume $\max_{x \in \mathcal{X}} f(x) = M$, and assume the observation error $\zeta_i = y_i - f(x_i)$ satisfies $|\zeta_i| < A$ for all $i$. Then we have the following error bound holds with probability at least $1 - n\varepsilon$:*

$$|\hat{\mu}_n(P_*) - \tilde{\mu}_n(P_*)| < (\frac{n}{\sigma^2} + \frac{n^2}{\sigma^4})(M + A)e_\varepsilon + O(e_\varepsilon^2) \tag{24}$$

$$|\hat{\sigma}_n^2(P_*) - \tilde{\sigma}_n^2(P_*)| < (1 + \frac{n}{\sigma^2})^2 e_\varepsilon + O(e_\varepsilon^2) \tag{25}$$

$$\left| \tilde{I}(\mathbf{y}_n; \hat{\mathbf{f}}_n | \{P_t\}_{t=1}^n) - \hat{I}(\mathbf{y}_n; \hat{\mathbf{f}}_n | \{P_t\}_{t=1}^n) \right| < \frac{n^{3/2}}{2\sigma^2} e_\varepsilon + O(e_\varepsilon^2) \tag{26}$$

*Proof.* Denote $e(P_*, Q) = \tilde{k}(P_*, Q) - \hat{k}(P_*, Q)$, $\mathbf{e}_n(P_*) = [e(P_*, P_1), \cdots, e(P_*.P_n)]^T$, and $[\mathbf{E}_n]_{i,j} = e(P_i, P_j)$. Now according to the matrix inverse perturbation expansion,

$$(X + \delta X)^{-1} = X^{-1} - X^{-1}\delta X X^{-1} + O(\|\delta X\|^2),$$

we have

$$(\hat{\mathbf{K}}_n + \sigma^2\mathbf{I} + \mathbf{E}_n)^{-1} = (\hat{\mathbf{K}}_n + \sigma^2\mathbf{I})^{-1} - (\hat{\mathbf{K}}_n + \sigma^2\mathbf{I})^{-1}\mathbf{E}_n(\hat{\mathbf{K}}_n + \sigma^2\mathbf{I})^{-1} + O(\|\mathbf{E}_n\|^2),$$

thus

$$
\begin{aligned}
\tilde{\mu}_n(P_*) =& (\hat{\mathbf{k}}_n(P_*) + \mathbf{e}_n(P_*))^T(\hat{\mathbf{K}}_n + \sigma^2\mathbf{I} + \mathbf{E}_n)^{-1}\mathbf{y}_n \\
=& \hat{\mu}_n(P_*) + \mathbf{e}_n(P_*)^T(\hat{\mathbf{K}}_n + \sigma^2\mathbf{I})^{-1}\mathbf{y}_n - \hat{\mathbf{k}}_n(P_*)^T(\hat{\mathbf{K}}_n + \sigma^2\mathbf{I})^{-1}\mathbf{E}_n(\hat{\mathbf{K}}_n + \sigma^2\mathbf{I})^{-1}\mathbf{y}_n \\
& + O(\|\mathbf{E}_n\|^2) + O(\|\mathbf{e}_n(P_*))\| \cdot \|\mathbf{E}_n\|) \\
\tilde{\sigma}_n^2(P_*) =& \hat{\sigma}_n^2(P_*) + e(P_*, P_*) - (\hat{\mathbf{k}}_n(P_*) + \mathbf{e}_n(P_*))^T(\hat{\mathbf{K}}_n + \sigma^2\mathbf{I} + \mathbf{E}_n)^{-1}(\hat{\mathbf{k}}_n(P_*) + \mathbf{e}_n(P_*)) \\
=& \hat{\sigma}_n^2(P_*) + e(P_*, P_*) - 2\mathbf{e}_n(P)^T(\hat{\mathbf{K}}_n + \sigma^2\mathbf{I})^{-1}\hat{\mathbf{k}}_n(P_*) \\
& + \hat{\mathbf{k}}_n(P)^T(\hat{\mathbf{K}}_n + \sigma^2\mathbf{I})^{-1}\mathbf{E}_n(\hat{\mathbf{K}}_n + \sigma^2\mathbf{I})^{-1}\hat{\mathbf{k}}_n(P_*) \\
& + O(\|\mathbf{E}_n\|^2) + O(\|\mathbf{e}_n\| \cdot \|\mathbf{E}_n\|) + O(\|\mathbf{e}_n\|^2 \cdot \|\mathbf{E}_n\|)
\end{aligned}
$$

Notic that the following holds with a probability at least $1 - n\varepsilon$, according to the Assumption 1,

$$|\mathbf{e}_n(P_*)^T(\hat{K}_n + \sigma^2\mathbf{I})^{-1}\mathbf{y}_n| \leq \|\mathbf{e}_n(P_*)\|_2\|(\hat{K}_n + \sigma^2\mathbf{I})^{-1}\|_2\|\mathbf{y}_n\|_2 \leq \frac{n}{\sigma^2}(M + A)e_\varepsilon,$$

$$
\begin{aligned}
|\hat{\mathbf{k}}_n(P_*)^T(\hat{\mathbf{K}}_n + \sigma^2\mathbf{I})^{-1}\mathbf{E}_n(\hat{\mathbf{K}}_n + \sigma^2\mathbf{I})^{-1}\mathbf{y}_n| &\leq \|\hat{\mathbf{k}}_n(P_*)\|_2\|(\hat{K}_n + \sigma^2\mathbf{I})^{-1}\|_2^2\|\mathbf{E}_n\|_2\|\mathbf{y}_n\|_2 \\
&\leq \sqrt{n}\sigma^{-4}ne_\varepsilon\sqrt{n}(M + A) = \frac{n^2}{\sigma^4}(M + A),
\end{aligned}
$$

here we use the fact that $\hat{K}_n$ semi-definite (which means $\|(\hat{K}_n + \sigma^2 I)^{-1}\|_2 \leq \sigma^{-2}$), $\hat{k}(P_*, P_*) \leq 1$, $|y_i| \leq M + A$. Combining these results, we have that

$$|\tilde{\mu}_n(P_*) - \hat{\mu}_n(P_*)| < (\frac{n}{\sigma^2} + \frac{n^2}{\sigma^4})(M + A)e_\varepsilon + O(e_\varepsilon^2),$$

holds with a probability at least $1 - n\varepsilon$.

Similarly, we can conduct the same estimation to $\mathbf{e}_n(P)^T(\hat{\mathbf{K}}_n + \sigma^2\mathbf{I})^{-1}\hat{\mathbf{k}}_n(P_*)$ and $\hat{\mathbf{k}}_n(P)^T(\hat{\mathbf{K}}_n + \sigma^2\mathbf{I})^{-1}\mathbf{E}_n(\hat{\mathbf{K}}_n + \sigma^2\mathbf{I})^{-1}\hat{\mathbf{k}}_n(P_*)$, and get

$$|\tilde{\sigma}_n^2(P_*) - \hat{\sigma}_n^2(P_*)| < (1 + \frac{n}{\sigma^2})^2 e_\varepsilon + O(e_\varepsilon^2)$$

holds with a probability at least $1 - n\varepsilon$.

It remains to estimate the error for estimating the information gain. Notice that, with a probability at least $1 - n\varepsilon$,

$$
\begin{aligned}
\left|\tilde{I}(\mathbf{y}_n; \hat{\mathbf{f}}_n | \{p_t\}_{t=1}^n) - \hat{I}(\mathbf{y}_n; \hat{\mathbf{f}}_n | \{p_t\}_{t=1}^n)\right| &= \left|\frac{1}{2}\log\frac{\det(\mathbf{I} + \sigma^{-2}\tilde{\mathbf{K}}_n)}{\det(\mathbf{I} + \sigma^{-2}\hat{\mathbf{K}}_n)}\right| \\
&= \left|\frac{1}{2}\log\det(\mathbf{I} - (\sigma^2\mathbf{I} + \hat{\mathbf{K}}_n)^{-1}\mathbf{E}_n)\right| \\
&= \left|\frac{1}{2}\mathrm{Tr}(\log(\mathbf{I} - (\sigma^2\mathbf{I} + \hat{\mathbf{K}}_n)^{-1}\mathbf{E}_n))\right| \\
&= \left|\frac{1}{2}\mathrm{Tr}(-(\sigma^2\mathbf{I} + \hat{\mathbf{K}}_n)^{-1}\mathbf{E}_n) + O(\|\mathbf{E}_n\|^2)\right| \\
&\leq \frac{n^{3/2}}{2\sigma^2}e_\varepsilon + O(\|\mathbf{E}_n\|^2),
\end{aligned}
$$

here the second equation uses the fact that $\det(AB^{-1}) = \det(A)\det(B)^{-1}$, and the third and fourth equations use $\log\det(I+A) = \operatorname{Tr}\log(I+A) = \operatorname{Tr}(A - \frac{A^2}{2} + \cdots)$. The last inequality follows from the fact

$$\operatorname{Tr}(\sigma^2\mathbf{I} + \hat{\mathbf{K}}_n)^{-1}\mathbf{E}_n) \leq \|(\sigma^2\mathbf{I} + \hat{\mathbf{K}}_n)^{-1}\|_F\|\mathbf{E}_n\|_F \leq n^{3/2}\sigma^{-2}e_\varepsilon$$

and $\hat{\mathbf{K}}_n$ is semi-definite. $\qquad\square$

With the uncertainty bound given by Lemma 3, let us prove that under inexact kernel estimations, the posterior mean is concentrated around the unknown reward function $\hat{f}$

**Theorem 4.** *Under the former setting as in Theorem 3, with probability at least $1 - \delta - n\varepsilon$, let $\sigma_\nu = \sqrt{\sigma_\zeta^2 + \sigma_E^2}$, taking $\sigma = 1 + \frac{2}{n}$, the following holds for all $x \in \mathcal{X}$:*

$$|\tilde{\mu}_n(P_x) - \hat{f}(P_x)| \leq \beta_n\tilde{\sigma}_n(P_x) + \beta_n(1+n)e_\varepsilon^{1/2} + (n+n^2)(M+A)e_\varepsilon, \tag{27}$$

$$\text{where } \beta_n = \left(b + \sigma_\nu\sqrt{2(\hat{\gamma}_n - \ln(\delta) + 1)}\right)$$

*Proof.* According to Lemma 2, equation (22), we have

$$|\hat{\mu}_n(P_x) - \hat{f}(P_x)| \leq \hat{\beta}_n\hat{\sigma}_n(P_x)$$

with

$$\hat{\beta}_n = b + \sigma_\nu\sqrt{2\left(\hat{I}(\mathbf{y}_n;\hat{\mathbf{f}}_n|\{P_t\}_{t=1}^n) + 1 + \ln(1/\delta)\right)} \leq \beta_n.$$

Notice that

$$|\tilde{\mu}_n(P_x) - \hat{f}(P_x)| \leq |\tilde{\mu}_n(P_x) - \hat{\mu}_n(P_x)| + |\hat{\mu}_n(P_x) - \hat{f}(P_x)|, \tag{28}$$

We also have (25), which means

$$\hat{\sigma}_n(P_x) = \sqrt{\hat{\sigma}_n(P_x)^2} \leq \sqrt{\tilde{\sigma}_n(P_x)^2 + (1+n)^2 e_\varepsilon} \leq \tilde{\sigma}_n(P_x) + (1+n)e_\varepsilon^{1/2}, \tag{29}$$

combining (24), (28) and (29), we finally get the result in (27).

$\qquad\square$

## B.3  Proofs for Theorem 1

Now we can prove our main theorem 1.

*Proof of Theorem 1.* Let $x^*$ maximize $\hat{f}(P_x)$ over $\mathcal{X}$. Observing that at each round $n \geq 1$, by the choice of $x_n$ to maximize the aquisition function $\tilde{\alpha}(x|\mathcal{D}_{n-1}) = \tilde{\mu}_{n-1}(P_x) + \beta_{n-1}\tilde{\sigma}_{n-1}(P_x)$, we have

$$\begin{aligned}
\tilde{r}_n &= \hat{f}(P_{x^*}) - \hat{f}(P_{x_n}) \\
&\leq \tilde{\mu}_{n-1}(P_{x^*}) + \beta_{n-1}\tilde{\sigma}_{n-1}(P_{x^*}) - \tilde{\mu}_{n-1}(P_{x_n}) + \beta_{n-1}\tilde{\sigma}_{n-1}(P_{x_n}) + 2Err(n-1, e_\varepsilon) \\
&\leq 2\beta_{n-1}\tilde{\sigma}_{n-1}(P_{x_n}) + 2Err(n-1, e_\varepsilon).
\end{aligned}$$

Here we denote $Err(n, e_\varepsilon) := \left(\beta_n(1+n) + \tilde{\sigma}_n(P_x)\sigma_\nu n^{3/4}\right)e_\varepsilon^{1/2} + (n+n^2)(M+A)e_\varepsilon$. The second inequality follows from (27),

$$\hat{f}(P_{x^*}) - \tilde{\mu}_{n-1}(P_{x^*}) \leq \beta_{n-1}\tilde{\sigma}_{n-1}(P_{x^*}) + Err(n-1, e_\varepsilon)$$

$$\tilde{\mu}_{n-1}(P_{x_n}) - \hat{f}(P_{x_n}) \leq \beta_{n-1}\tilde{\sigma}_{n-1}(P_{x_n}) + Err(n-1, e_\varepsilon),$$

and the third inequality follows from the choice of $x_n$:

$$\tilde{\mu}_{n-1}(P_{x^*}) + \beta_{n-1}\tilde{\sigma}_{n-1}(P_{x^*}) \leq \tilde{\mu}_{n-1}(P_{x_n}) + \beta_{n-1}\tilde{\sigma}_{n-1}(P_{x_n}).$$

Thus we have

$$\tilde{R}_n = \sum_{t=1}^{n} \tilde{r}_t \leq 2\beta_n \sum_{t=1}^{n} \tilde{\sigma}_{t-1}(P_{x_t}) + \sum_{t=1}^{T} Err(t-1, e_\varepsilon).$$

From Lemma 4 in [9], we have that

$$\sum_{t=1}^{n} \tilde{\sigma}_{t-1}(P_{x_t}) \leq \sqrt{4(n+2)\ln\det(I + \sigma^{-2}\tilde{K}_n)} \leq \sqrt{4(n+2)(\hat{\gamma}_n + \frac{n^{\frac{3}{2}}}{2}e_\varepsilon)},$$

and thus

$$2\beta_n \sum_{t=1}^{n} \tilde{\sigma}_{t-1}(P_{x_t}) = O\left(\sqrt{n\hat{\gamma}_n} + \sqrt{n\hat{\gamma}_n(\hat{\gamma}_n - \ln\delta)} + \sqrt{n^{\frac{5}{2}}e_\varepsilon} + \sqrt{n^{\frac{5}{2}}(\hat{\gamma}_n - \ln\delta)e_\varepsilon}\right). \quad (30)$$

On the other hand, notice that

$$\sum_{t=1}^{n} Err(t-1, e_\varepsilon) = O\left(n^2\sqrt{(\hat{\gamma}_n - \ln\delta)e_\varepsilon} + (n^2 + n^3)e_\epsilon\right), \quad (31)$$

we find that the $e_\varepsilon$ term in (30) can be controlled by in (31), thus we immediately get the result. □

### B.4 Proofs for Theorem 2

*Proof.* Define the square of the MMD distance between $P_{x_1}, P_{x_2}$ as $d_M(x_1, x_2)$, we have

$$d_M(x_1, x_2)$$
$$= \int_{\mathbb{R}^d} k(x, x')P_{x_1}(x)P_{x_1}(x')\mathrm{d}x\mathrm{d}x' + \int_{\mathbb{R}^d} k(x, x')P_{x_2}(x)P_{x_2}(x')\mathrm{d}x\mathrm{d}x'$$
$$- 2\int_{\mathbb{R}^d} k(x, x')P_{x_1}(x)P_{x_2}(x')\mathrm{d}x\mathrm{d}x'$$
$$= \int_{\mathbb{R}^d} (k(x - x_1, x' - x_1) + k(x - x_2, x' - x_2) - 2k(x - x_1, x' - x_2))P_0(x)P_0(x')\mathrm{d}x\mathrm{d}x'.$$

It is not hard to verify that $d_M$ is shift invariant: $d_M(x_1, x_2) = d_M(x_1 - x_2, 0)$, and $d_M$ has $r$-th bounded derivatives, thus $\hat{k}^*(x_1, x_2) := \hat{k}(P_{x_1}, P_{x_2}) = \exp(-\alpha d_M(x_1, x_2))$ is shift invariant with $r$-th bounded derivatives. Then take $\mu(x)$ as the Lebesgue measure over $\mathcal{X}$, according to Theorem 4, [17], the integral operator $T_{k,\mu} : T_{k,\mu}f(x) = \int_{\mathcal{X}} K(x, y)f(y)d\mu(y)$ is a symmetric compact operator in $L_2(\mathcal{X}, \mu)$, and the spectrum of $T_{k,\mu}$ satisfies

$$\lambda_n(T_{k,\mu}) = O(n^{-1-r/d}).$$

Then according to Theorem 5 in [30], we have $\hat{\gamma}_n = O(n^{\frac{d(d+1)}{r+d(d+1)}}\log(n))$, which finish the proof. □

## C  Evaluation Details

### C.1  Implementation

In our implementation of AIRBO, we design the kernel $k$ used for MMD estimation to be a linear combination of multiple Rational Quadratic kernels as its long tail behavior circumvents the fast decay issue of kernel [6]:

$$k(x, x') = \sum_{a_i \in \{0.2, 0.5, 1, 2, 5\}} \left(1 + \frac{(x - x')^2}{2a_i l_i^2}\right)^{-a_i}, \quad (32)$$

where $l_i$ is a learnable lengthscale and $a_i$ determines the relative weighting of large-scale and small-scale variations.

Depending on the form of input distributions, the sampling and sub-sampling sizes for Nyström MMD estimator are empirically selected via experiments. Moreover, as the input uncertainty is already modeled in the surrogate, we employ a classic UCB-based acquisition as Eq. 5 with $\beta = 2.0$ and maximize it via an L-BFGS-B optimizer.

## D  More Experiments

### D.1  Comparing with the Other Models

To compare the modeling performances with the other models, we design the input uncertainty to follow a step-changing Chi-squared distribution: $P_x = \chi^2(g(x), \sigma = 0.01)$, where $g(x) = 0.5$ if $x \in [0.0, 0.6)$ and $g(x) = 7.0$ when $x \in [0.6, 1.0]$. Due to this sudden parameter change, the uncertainty at point $x = 0.6$ is expected to be asymmetric: 1) on its left-hand side, as the Chi-squared distribution becomes quite lean and sharp with a small value of $g(x) = 0.5$, the distance from $x = 0.6$ to its LHS points, $x_{lhs} \in [0.0, 0.6)$, are relatively large, thus their covariances are small, resulting a fast-growing uncertainty. 2)Meanwhile, when $x \in [0.6, 1.0]$, the $g(x)$ suddenly increases to 7.0, rendering the input distribution a quite flat one with a long tail. Therefore, the distances between $x = 0.6$ and its RHS points become relatively small, which leads to large covariances and small uncertainties for points in $[0.6, 1.0]$. As a result, we expect to observe an asymmetric posterior uncertainty at $x = 0.6$.

Several surrogate models are employed in this comparison, including:

- **MMDGP-nystrom(160/10)** is our method with a sampling size $m = 160$ and sub-sampling size $h = 10$. Its complexity is $O(MNmh)$, where $M$ and $N$ are the sizes of training and test samples (Note: all the models in this experiment use the same training and testing samples for a fair comparison).

- **uGP(40)** is the surrogate from [25], which employs an integral kernel with sampling size $m = 40$. Due to its $O(MNm^2)$ complexity, we set the sampling size $m = 40$ to ensure a similar complexity as ours.

- **uGP(160)** is also the surrogate from [25] but uses a much larger sampling size ($m = 160$). Given the same training and testing samples, its complexity is 16 times higher than **MMDGP-nystrom(160/10)**.

- **skl** is a robust GP surrogate equipped with a symmetric KL-based kernel, which is described in [20].

- **ERBF** [13] assumes the input uncertainty to be Gaussians and employs a close-form expected RBF kernel.

- **GP** utilizes a noisy Gaussian Process model with a learnable output noise level.

According to Figure 7a, our method, MMDGP-nystrom(160/10), can comprehensively quantify the sudden change of the input uncertainty, evidenced by its abrupt posterior change at $x = 0.6$. However, Figure 7b shows that uGP(40) with the same complexity fails to model the uncertainty correctly. We suspect this is because uGP requires much larger samples to stabilize its estimation of the integral kernel and thus can perform poorly with insufficient sample size, so we further evaluate the uGP(160) with a much larger sampling size $m = 160$ in Figure 7c. It does successfully alleviate the issue but also results in a 16 times higher complexity. Apart from this, Figure 7d suggests the noisy GP model with a learnable output noise level is not aware of this uncertainty change at all as it treats the inputs as the exact values instead of random variables. Moreover, Figure 7e and 7f show that both the skl and ERBF fail in this case, this may be due to their misassumption of Gaussian input uncertainty.

### D.2  Ablation Test for Nyström Approximation

In this experiment, we aim to examine the effect of Nyström approximation for optimization. To this end, we choose to optimize an RKHS function (Figure 4a) under a beta input distribution: $P_x = beta(\alpha = 0.4, \beta = 0.2, \sigma = 0.1)$. Several amortized candidates include:

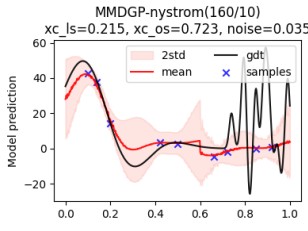

(a) MMD-GP with a Nystrom estimator, in which the sampling size $m = 160$ and sub-sampling size $h = 10$.

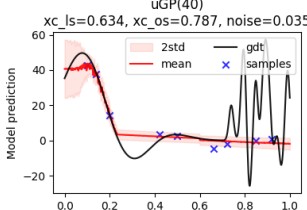

(b) uGP model that uses an integral kernel [26] and a sampling size of $m = 40$.

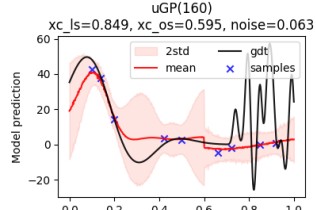

(c) uGP with an integral kernel [26] and uses a much larger sampling size ($m = 160$).

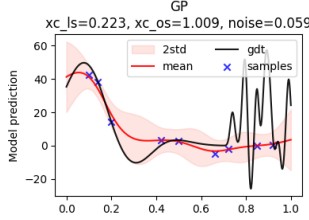

(d) Conventional noisy GP model.

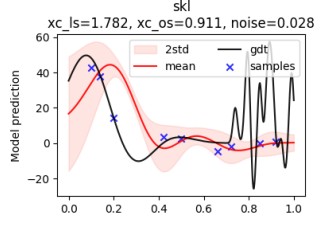

(e) GP model with a symmetric KL-divergence kernel [20].

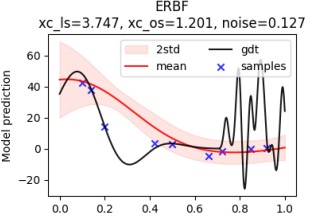

(f) Robust GP model with an expected RBF kernel [13]

Figure 7: Modeling performance with a step-changing Chi-squared distribution.

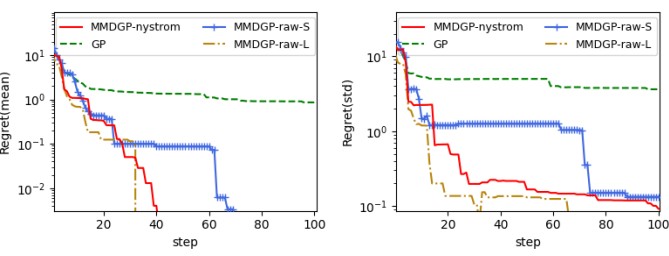

Figure 8: Ablation test for the Nyström approximation.

- *MMDGP-nystrom* is our method with Nystrom approximation, in which the sampling size $m = 16$ and sub-sampling size $h = 9$. Its complexity is $O(MNmh)$, where $M$ and $N$ are the sizes of training and test samples respectively, $m$ is the sampling size for MMD estimation, and $h$ indicates the sub-sampling size during the Nystrom approximation.

- *MMDGP-raw-S* does not use the Nystrom approximation but employs an empirical MMD estimator. Due to its $O(MNm^2)$ complexity, we set the sampling size $m = 12$ to ensure a similar complexity as the *MMDGP-nystrom*.

- *MMDGP-raw-L* also uses an empirical MMD estimator, but with a larger sampling size ($m = 16$).

- *GP* utilizes a vanilla GP with a learnable output noise level and optimizes with the upper-confidence-bound acquisition[2].

According to Figure 8, we find that 1) with sufficient computation power, the *MMDGP-raw-L* can obtain the best performance by using a large sample size. 2)However, with limited complexity, the performance *MMDGP-raw-S* degrades obviously while the *MMDGP-nystrom* performs much better. This suggests that the Nyström approximation can significantly improve the efficiency with a mild cost of performance degradation. 3) All the MMDGP-based methods are better than the vanilla *GP-UCB*.

---

[2]For a fair comparison, all the methods in this test use a UCB acquisition.

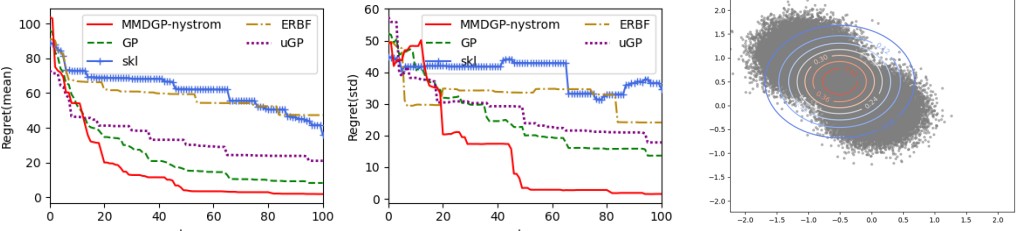

Figure 9: Optimization regret on 10D bumped-bowl problem.

Figure 10: The input GMM distribution.

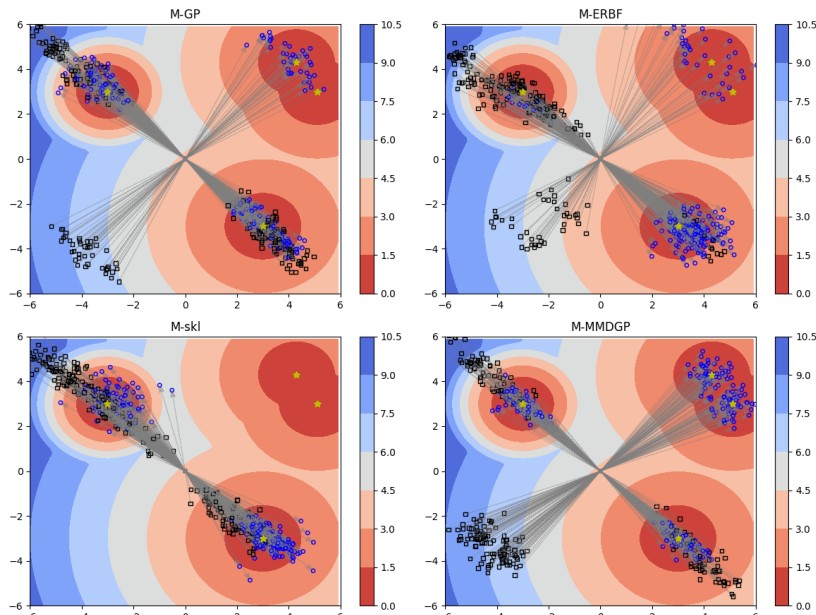

Figure 11: Simulation results of the push configurations found by different algorithms.

### D.3 Optimization on 10D Bumped-Bowl Problem

To further evaluate AIRBO's optimization performance on the high-dimensional problem, we employ a 10-dimensional bumped bowl function from [27, 19]:

$$f(\boldsymbol{x}) = g(\boldsymbol{x}_{1:2})h(\boldsymbol{x}_{3:}), \text{where} \begin{cases} g(\boldsymbol{x}) = 2\log\left(0.8\|\boldsymbol{x}\|^2 + e^{-10\|\boldsymbol{x}\|^2}\right) + 2.54 \\ h(\boldsymbol{x}) = \sum_i^d 5x_i^2 + 1 \end{cases} \tag{33}$$

Here, $x_i$ is the i-th dimension of $\boldsymbol{x}$, $\boldsymbol{x}_{1:2}$ represents the first 2 dimensions for the variable, and $\boldsymbol{x}_{3:}$ indicates the rest dimensions. The input uncertainty is designed to follow a concatenated distribution of a 2D circular distribution($r = 0.5$) and a multivariate normal distribution with a zero mean and diagonal covariance of 0.01.

Figure 9 shows the mean and std values of the optimization regrets. We note that 1)when it comes to a high-dimensional problem and complex input distribution, the misassumption of Gaussian input uncertainty renders the *skl* and *ERBF* fail to locate the robust optimum and get stuck at local optimums. 2)Our method outperforms the others and can find the robust optimum efficiently and stably, while the *uGP* with a similar inference cost suffers the instability caused by insufficient sampling and stumbles over iterations, which can be evidenced by its high std values of optimization regret.

## D.4 Robust Robot Pushing

This benchmark is based on a Box2D simulator from [31], where our objective is to identify a robust push configuration, enabling a robot to push a ball to predetermined targets under input randomness. In our experiment, we simplify the task by setting the push angle to $r_a = \arctan \frac{r_y}{r_x}$, ensuring the robot is always facing the ball. Also, we intentionally define the input distribution as a two-component Gaussian Mixture Model as follows:

$$(r_x, r_y, r_t) \sim GMM\left(\mu = \begin{bmatrix} 0 & 0 & 0 \\ -1 & 1 & 0 \end{bmatrix}, \Sigma = \begin{bmatrix} 0.1^2 & -0.3^2 & 1e-6 \\ -0.3^2 & 0.1^2 & 1e-6 \\ 1e-6 & 1e-6 & 1.0^2 \end{bmatrix}, w = \begin{bmatrix} 0.5 \\ 0.5 \end{bmatrix}\right), \quad (34)$$

where the covariance matrix $\Sigma$ is shared among components and $w$ is the weights of mixture components. Meanwhile, as the SKL-UCB and ERBF-UCB surrogates can only accept Gaussian input distributions, we choose to approximate the true input distribution with a Gaussian. As shown in Figure 10, the approximation error is obvious, which explains the performance gap among these algorithms in Figure 5b.

Apart from the statistics of the found pre-images in Figure 6, we also simulate the robot pushes according to the found configurations and visualize the results in Figure 11. In this figure, each black hollow square represents an instance of the robot's initial location, the grey arrow indicates the push direction and duration, and the blue circle marks the ball's ending position after the push. We can find that, as the GP-UCB ignores the input uncertainty, it randomly pushes to these targets and the ball ending positions fluctuate. Also, due to the incorrect assumption of the input distribution, the SKL-UCB and ERBF-UCB fail to control the ball's ending position under input randomness. On the contrary, AIRBO successfully recognizes the twin targets in quadrant I as an optimal choice and frequently pushes to this area. Moreover, all the ball's ending positions are well controlled and centralized around the targets under input randomness.

