## A Nyström Estimator Error bound

Nyström estimator can easily approximate the kernel mean embedding $\psi_{p_1}, \psi_{p_2}$ as well as the MMD distance between two distribution density $p_1$ and $p_2$. We need first assume the boundedness of the feature map to the kernel $k$:

**Assumption 2.** *There exists a positive constant $K \leq \infty$ such that $\sup_{x \in \mathcal{X}} \|\phi(x)\| \leq K$*

The true MMD distance between $p_1$ and $p_2$ is denoted as $\mathrm{MMD}(p_1, p_2)$. The estimated MMD distance when using a Nyström sample size $n_i$, sub-sample size $m_i$ for $p_i$ respectively, is denoted as $\mathrm{MMD}_{(p_i, m_i, n_i)}$. Then the error

$$\mathrm{Err}_{(p_i, n_i, m_i)} := |\mathrm{MMD}(p_1, p_2) - \mathrm{MMD}_{(p_i, m_i, n_i)}|$$

and now we have the lemma from Theorem 5.1 in [8]

**Lemma 1.** *Let Assumption 2 hold. Furthermore, assume that for $i \in 1, 2$, the data points $X_1^i, \cdots, X_{n_i}^i$ are drawn i.i.d. from the distribution $\rho_i$ and that $m_i \leq n_i$ sub-samples $\tilde{X}_1^i, \cdots, \tilde{X}_{m_i}^i$ are drawn uniformly with replacement from the dataset $\{X_1^i, \cdots, X_{n_i}^i\}$. Then, for any $\delta \in (0, 1)$, it holds with probability at least $1 - 2\delta$*

$$Err_{(p_i, n_i, m_i)} \leq \sum_{i=1,2} \left( \frac{c_1}{\sqrt{n_i}} + \frac{c_2}{m_i} + \frac{\sqrt{\log(m_i/\delta)}}{m_i} \sqrt{\mathcal{N}^{p_i}\left(\frac{12K^2 \log(m_i/\delta)}{m_i}\right)} \right),$$

*provided that, for $i \in \{1, 2\}$,*

$$m_i \geq \max(67, 12K^2 \|C_i\|_{\mathcal{L}(\mathcal{H})}^{-1}) \log(m_i/\delta)$$

*where $c_1 = 2K\sqrt{2\log(6/\delta)}, c_2 = 4\sqrt{3}K \log(12/\delta)$ and $c_4 = 6K\sqrt{\log(12/\delta)}$. The notation $\mathcal{N}^{p_i}$ denotes the effective dimension associated to the distribution $p_k$.*

*Specifically, when the effective dimension $\mathcal{N}$ satisfies, for some $c \geq 0$,*

- *either $\mathcal{N}^{\rho_i}(\sigma^2) \leq c\sigma^{2-\gamma}$ for some $\gamma \in (0, 1)$,*

- *or $\mathcal{N}^{\rho_i}(\sigma^2) \leq \log(1 + c/\sigma^2)/\beta$, for some $\beta > 0$.*

*Then, choosing the subsample size $m$ to be*

- *$m_i = n_i^{1/(2-\gamma)} \log(n_i/\delta)$ in the first case*

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

$$where\ \tilde{\beta}_n = \left(b + \sigma_\nu\sqrt{2(\tilde{I}(\mathbf{y}_n; \hat{\mathbf{f}}_n|\{P_t\}_{t=1}^n) - \ln(\delta) + 1)}\right) \tag{46}$$

*Proof.* According to Lemma 2, equation (25), we have

$$|\hat{\mu}_n(P_x) - \hat{f}(P_x)| \leq \hat{\beta}_n\hat{\sigma}_n(P_x) \tag{47}$$

with

$$\hat{\beta}_n = b + \sigma_\nu\sqrt{2\left(\hat{I}(\mathbf{y}_n; \hat{\mathbf{f}}_n|\{P_t\}_{t=1}^n) + 1 + \ln(1/\delta)\right)}. \tag{48}$$

Notice that

$$|\tilde{\mu}_n(P_x) - \hat{f}(P_x)| \leq |\tilde{\mu}_n(P_x) - \hat{\mu}_n(P_x)| + |\hat{\mu}_n(P_x) - \hat{f}(P_x)|, \tag{49}$$

$$\hat{\beta}_n = b + \sigma_\nu\sqrt{2\left(\hat{I}(\mathbf{y}_n; \hat{\mathbf{f}}_n|\{P_t\}_{t=1}^n) + 1 + \ln(1/\delta)\right)} \tag{50}$$

$$\leq b + \sigma_\nu\sqrt{2\left(\tilde{I}(\mathbf{y}_n; \hat{\mathbf{f}}_n|\{P_t\}_{t=1}^n) + \frac{n^{3/2}}{2}e_\varepsilon + 1 + \ln(1/\delta)\right)} \tag{51}$$

$$\leq b + \sigma_\nu\sqrt{2\left(\tilde{I}(\mathbf{y}_n; \hat{\mathbf{f}}_n|\{P_t\}_{t=1}^n) + 1 + \ln(1/\delta)\right)} + \sigma_\nu n^{3/4}e_\varepsilon^{1/2} \tag{52}$$

where the second inequality follows from Theorem 2, (30), and the third inequality follows from the inequality $\sqrt{a_1 + a_2} \leq \sqrt{a_1} + \sqrt{a_2}, a_1 > 0, a_2 > 0$.

We also have (29), which means

$$\hat{\sigma}_n(P_x) = \sqrt{\hat{\sigma}_n(P_x)^2} \leq \sqrt{\tilde{\sigma}_n(P_x)^2 + (1 + n)^2 e_\varepsilon} \leq \tilde{\sigma}_n(P_x) + (1 + n)e_\varepsilon^{1/2}, \tag{53}$$

combining (28), (49), (50) and (53), we finally get the result in (45).

$\qquad\square$

## B.3 Proofs for Theorem 1

Now we can prove our main theorem 1.

*Proof of Theorem 1.* Let $x^*$ maximize $\hat{f}(P_x)$ over $\mathcal{X}$. Observing that at each round $n \geq 1$, by the choice of $x_n$ to maximize the aquisition function $\tilde{\alpha}(x|\mathcal{D}_{n-1}) = \tilde{\mu}_{n-1}(P_x) + \tilde{\beta}_{n-1}\tilde{\sigma}_{n-1}(P_x)$, we have

$$\tilde{r}_n = \hat{f}(P_{x^*}) - \hat{f}(P_{x_n}) \tag{54}$$

$$\leq \tilde{\mu}_{n-1}(P_{x^*}) + \tilde{\beta}_{n-1}\tilde{\sigma}_{n-1}(P_{x^*}) - \tilde{\mu}_{n-1}(P_{x_n}) + \tilde{\beta}_{n-1}\tilde{\sigma}_{n-1}(P_{x_n}) + 2Err(n-1,e_\varepsilon) \tag{55}$$

$$\leq 2\tilde{\beta}_{n-1}\tilde{\sigma}_{n-1}(P_{x_n}) + 2Err(n-1,e_\varepsilon). \tag{56}$$

Here we denote $Err(n,e_\varepsilon) := \left(\tilde{\beta}_n(1+n) + \tilde{\sigma}_n(P_x)\sigma_\nu n^{3/4}\right)e_\varepsilon^{1/2} + \left(n+n^2\right)(M+A)e_x$. The second inequality follows from (45),

$$\hat{f}(P_{x^*}) - \tilde{\mu}_{n-1}(P_{x^*}) \leq \tilde{\beta}_{n-1}\tilde{\sigma}_{n-1}(P_{x^*}) + Err(n-1,e_\varepsilon) \tag{57}$$

$$\tilde{\mu}_{n-1}(P_{x_n}) - \hat{f}(P_{x_n}) \leq \tilde{\beta}_{n-1}\tilde{\sigma}_{n-1}(P_{x_n}) + Err(n-1,e_\varepsilon), \tag{58}$$

and the third inequality follows from the choice of $x_n$:

$$\tilde{\mu}_{n-1}(P_{x^*}) + \tilde{\beta}_{n-1}\tilde{\sigma}_{n-1}(P_{x^*}) \leq \tilde{\mu}_{n-1}(P_{x_n}) + \tilde{\beta}_{n-1}\tilde{\sigma}_{n-1}(P_{x_n}).$$

Thus we have

$$\tilde{R}_n = \sum_{t=1}^{n} \tilde{r}_t \leq 2\tilde{\beta}_n \sum_{t=1}^{n} \tilde{\sigma}_{t-1}(P_{x_t}) + \sum_{t=1}^{T} Err(t-1,e_\varepsilon). \tag{59}$$

From Lemma 4 in [9], we have that

$$\sum_{t=1}^{n} \tilde{\sigma}_{t-1}(P_{x_t}) \leq \sqrt{4(n+2)\ln\det(I+\sigma^{-2}\tilde{K}_n)} \leq \sqrt{4(n+2)\tilde{\gamma}_n},$$

and thus

$$2\tilde{\beta}_n \sum_{t=1}^{n} \tilde{\sigma}_{t-1}(P_{x_t}) = O\left(\sqrt{n\tilde{\gamma}_n} + \sqrt{n\tilde{\gamma}_n(\tilde{\gamma}_n - \ln\delta)}\right).$$

On the other hand, notice that

$$\sum_{t=1}^{n} Err(t-1,e_\varepsilon) = O\left((\sqrt{\tilde{\gamma}_n}n^2 + n^{7/4})e_\varepsilon + (n^2+n^3)e_\epsilon\right),$$

we immediately get the result. $\qquad\square$

## C  Evaluation Details

### C.1  Implementation

In our implementation of AIRBO, we design the kernel $k$ used for MMD estimation to be a linear combination of multiple Rational Quadratic kernels as its long tail behavior circumvents the fast decay issue of kernel [6]:

$$k(x,x') = \sum_{a_i \in \{0.2,0.5,1,2,5\}} \left(1 + \frac{(x-x')^2}{2a_i l_i^2}\right)^{-a_i}, \tag{60}$$

where $l_i$ is a learnable lengthscale and $a_i$ determines the relative weighting of large-scale and small-scale variations.

Depending on the form of input distributions, the sampling and sub-sampling sizes for Nyström MMD estimator are empirically selected via experiments. Moreover, as the input uncertainty is already modeled in the surrogate, we employ a classic UCB-based acquisition as Eq. 5 with $\beta = 2.0$ and maximize it via an L-BFGS-B optimizer.

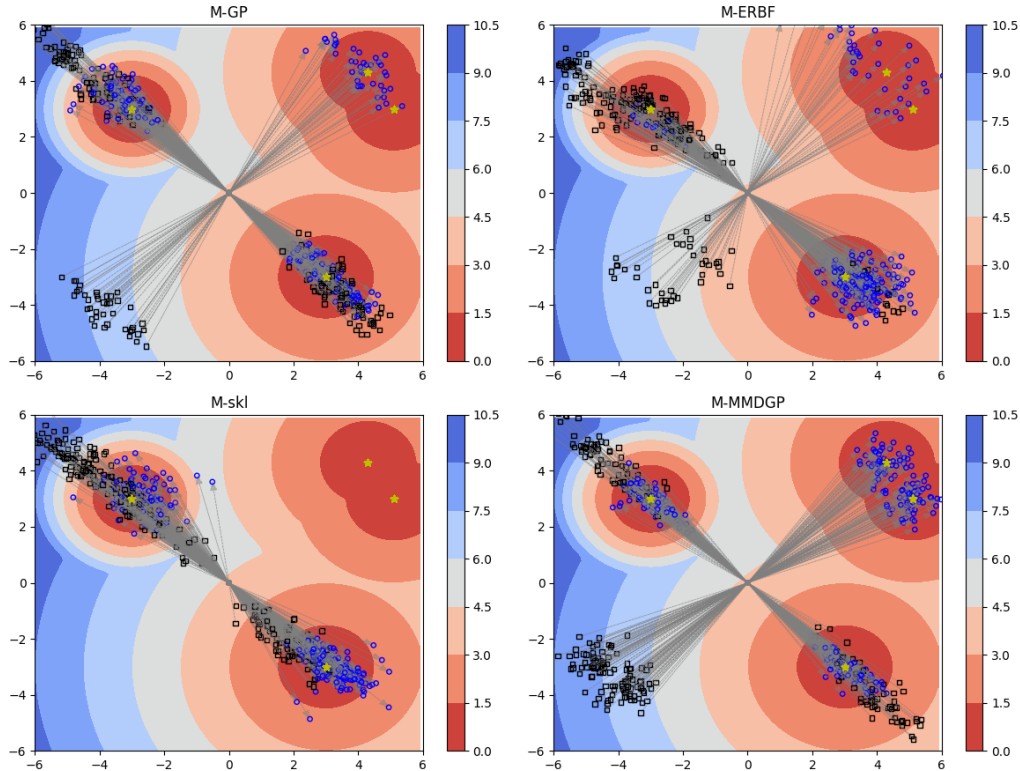

Figure 7: Simulation results of the push configurations found by different algorithms.

499 ## C.2  Supplementary Experiments

**Robust Robot Pushing:** This benchmark is based on a Box2D simulator from [30], where our objective is to identify a robust push configuration, enabling a robot to push a ball to predetermined targets under input randomness. In our experiment, we simplify the task by setting the push angle to $r_a = \arctan \frac{r_y}{r_x}$, ensuring the robot is always facing the ball. Also, we intentionally define the input distribution as a two-component Gaussian Mixture Model as follows:

$$(r_x, r_y, r_t) \sim GMM\Big(\mu = \begin{bmatrix} 0 & 0 & 0 \\ -1 & 1 & 0 \end{bmatrix}, \mathbf{\Sigma} = \begin{bmatrix} 0.1^2 & -0.3^2 & 1e-6 \\ -0.3^2 & 0.1^2 & 1e-6 \\ 1e-6 & 1e-6 & 1.0^2 \end{bmatrix}, w = \begin{bmatrix} 0.5 \\ 0.5 \end{bmatrix}\Big),$$

where the covariance matrix $\Sigma$ is shared among components and $w$ is the weights of mixture components. Figure 5b shows some example samples from this GMM distribution. Meanwhile, as the SKL-UCB and ERBF-UCB surrogates can only accept Gaussian input distributions, we choose to approximate the true input distribution with a Gaussian. As shown in Figure 5b, the approximation error is obvious, which explains the performance gap among these algorithms in Figure 5c.

Apart from the statistics of the found pre-images in Figure 6, we also simulate the robot pushes according to the found configurations and visualize the results in Figure 7. In this figure, each black hollow square represents an instance of the robot's initial location, the grey arrow indicates the push direction and duration, and the blue circle marks the ball's ending position after the push. We can find that, as the GP-UCB ignores the input uncertainty, it randomly pushes to these targets and the ball ending positions fluctuate. Also, due to the incorrect assumption of the input distribution, the SKL-UCB and ERBF-UCB fail to control the ball's ending position under input randomness. On the contrary, AIRBO successfully recognizes the twin targets in quadrant I as an optimal choice and frequently pushes to this area. Moreover, all the ball's ending positions are well controlled and centralized around the targets under input randomness.