# OpenReview forum: "Efficient Robust Bayesian Optimization for Arbitrary Uncertain inputs"
_NeurIPS.cc/2023/Conference — NeurIPS 2023 poster_

### Official Review · Reviewer_UjwV · 2023-07-03

**Soundness:** 2 fair
**Presentation:** 1 poor
**Contribution:** 3 good
**Rating:** 5
**Confidence:** 3

**Summary:**

The paper proposes a new method for Bayesian optimization under uncertain inputs, where the distribution of an input can be complex and unknown but can be sampled from. The paper proposes an MMD based kernel between probability distributions and the use of the Nystrom approximation to make the computations tractable. A UCB-type algorithm is used. The paper provides regret bounds accounting for the approximation error due to sampling / Nystrom approximation, and compares the algorithm to some baselines.

**Strengths:**

1. The motivation of requiring to do BO with uncertain inputs with complex distributions that may not be known in closed form is compelling.
2. The regret bound incorporating approximation error and the corresponding implications on the sampling size $m$ are useful and interesting.

In general I believe the work has potential given that the technical issues raised in the Weaknesses section are sufficiently addressed.

**Weaknesses:**

1. **Validity of kernel**. The paper designs an MMD-based kernel between probability distributions $\hat k = \eta(\text{MMD}(P, Q))$. When designing a new kernel, it is important to prove that it is a valid kernel. However, no such proof is given (or references to previous literature that does something similar). $\eta$ is said to be a "normalizing function with range $[0, 1]$". What conditions does $\eta$ need to fulfill in order for $\hat k$ to be a valid kernel?

2. **Empirical evaluation**. The empirical evaluation is unconvincing for the following reasons:
* The Nystrom approximation is adopted for efficiency. It is important to measure how much using this approximation affects performance beyond simply a qualitative comparison by looking at the posteriors in Figure 3. The empirical evaluation should do an ablation study that includes MMDGP without the Nystrom approximation, at a sampling size that results in the same inference time as that with the Nystrom approximation. What if removing the Nystrom approximation results in better performance with no decrease in efficiency?
* The closest work seems to be Oliveira et. al. (2019). Why isn't their algorithm one of the baselines?
* The error bars are simply too large and overlap too much. Any performance improvement could be due to randomness; please run with more trials to decrease the error bars so that the performances can be meaningfully compared.

3. **Writing**. The writing has much to be improved:
* Line 174 "One important theoretical guarantee to conduct GP model..." and in Theorem 1 "...running Gaussian Process with acquisition function..." I believe you mean "Bayesian optimization" in these contexts instead of "Gaussian process". The GP is the model, the algorithm is BO which relies on a GP. The thing being run is the algorithm, not the model.
* Line 151 "sampling size $N$" do you mean $m$?
* Many grammatical and typo errors: the last word in the title is not capitalized; line 178 "For $\hat k$ be radial kernels" and "For $\hat k$ be linear kernel", should be, for example, "If $\hat k$ is a radial kernel"; Lines 201 and 205 are missing periods; Line 256 "can be more pronounced impact" should be "can have".
* [24] and [25] point to the same reference.

**Questions:**

No additional questions.

**Limitations:**

Yes.

---

> ### Author Rebuttal · Authors · 2023-08-07
>
> We sincerely thank the reviewer for the insightful comments, especially regarding the constructive suggestions on the kernel validity, ablation test, and experiment.
>
> ### 1. Validity of kernel
>
> >  When designing a new kernel, it is important to prove that it is a valid kernel.
> >  $\eta$ is said to be a "normalizing function with range [0, 1]". What conditions does $\eta $ need to fulfill for $\hat{k}$ to be a valid kernel?
>
> Thanks for pointing this out! In our paper, we focus on the case that $\eta(x) = \exp{(-\alpha x)}$ and $\hat{k} = \exp(-\alpha \text{MMD}(P,Q; k))$. It can be proved to be a valid kernel by using the Theorem 2.2 from [1]: for any type of bi-functions defined on $\mathcal{P}\times\mathcal{P}$ as $\hat k(P, Q) = \sum_{i=1}^{\infty} a_i  \langle \psi_P, \psi_Q \rangle_k$ with $a_i \ge 0$, $\psi$ is the kernel mean map from $\mathcal{P}$ to $\mathcal{H}_k$, then $\hat k(P, Q)$ is a valid kernel; In addition if $a_i > 0$, this kernel is universal over $C(\mathcal{P})$, given the mean map $\psi$ is injective and the space of $P, Q$ defined on is compact. In this case, an MMD kernel with RBF mapping $\hat k(P, Q) = \exp(-\alpha \Vert \psi_P - \psi_Q \Vert_k^2)$ is ensured to be valid and universal.
>
> ### 2. Ablation test for Nystrom approximation
>
> > The Nystrom approximation is adopted for efficiency. It is important to measure how much using this approximation affects performance.
> > The empirical evaluation should do an ablation study that includes MMDGP without the Nystrom approximation, at a sampling size that results in the same inference time as that with the Nystrom approximation.
>
> Thanks for this constructive suggestion! We have conducted an ablation test for the Nystrom approximation. In this experiment, we employ an RKHS function and set the input uncertainty to follow a beta distribution (see Sec 5.2 in the paper). Several candidates are to study the effect of Nystrom approximation:
>
> + *MMDGP-nystrom* is our method with Nystrom approximation, in which the sampling size $m=16$ and sub-sampling size $h=9$. Its complexity is $O(MNmh)$, where $M$ and $N$ are the sizes of training and test samples respectively, $m$ is the sampling size for MMD estimation, and $h$ indicates the sub-sampling size during the Nystrom approximation.
> + *MMDGP-raw-S* does not use the Nystrom approximation but employs an empirical MMD estimator. Due to its $O(MNm^2)$ complexity, we set the sampling size $m=12$ to ensure a similar complexity as the *MMDGP-nystrom*.
> + *MMDGP-raw-L* also uses an empirical MMD estimator, but with a larger sampling size ($m = 16$).
> + *GP* utilizes a vanilla GP with a learnable output noise level and optimizes with the upper-confidence-bound acquisition [^a].
>
> Figure 7a in the supplementary_results.pdf shows that i) with sufficient computation power, the *MMDGP-raw-L* can obtain the best performance by using a large sample size.  However, ii) with the same inference complexity, the *MMDGP-nystrom* performs much better than the *MMDGP-raw-S*, suggesting the Nystrom approximation can significantly improve the efficiency with a mild cost of performance degradation. iii) All the MMDGP-based methods are better than the vanilla GP-UCB.
>
> ### 3. Experiments
>
> > The closest work seems to be Oliveira et. al. (2019). Why isn't their algorithm one of the baselines?
> > The error bars are simply too large and overlap too much. please run with more trials to decrease the error bars so that the performances can be meaningfully compared.
>
> Thanks for pointing this out! We have cited this paper and included the uGP-UCB method in our baselines and rerun all the experiments with more trials to provide a  statistically-meaningful comparison.
>
> The uGP-UCB method in [2] employs an integral kernel over probability measures $\mathcal{P}$: $\hat{k}(P, P^\prime):= \int_\mathcal{X} \int_\mathcal{X} k(x, x^\prime) dP(x) dP^\prime(x^\prime)$. Since it does not mention how to compute this kernel and no public code is available, we compute this integral kernel via sampling, resulting in an inference complexity of $O(MNm^2)$. Here $M$ and $N$ are the sizes of training and test samples respectively, and $m$ is the number of samples used for estimating the integral.
>
> In the supplementary results, Figure 9a compares our method with the baselines on an RKHS function (Figure 4a in the paper) with a Gaussian input uncertainty. All the robust methods perform well except the vanilla GP-UCB stacks to a local optimum. Also, we notice that the *uGP* method performs slightly better than the others in this case, but we will see later that our method outperforms in more complex distributions and high-dimensional cases.
>
> Figure 9b further compares these models on a double-peak function (Figure 4c in the paper) and a beta distribution. In this case, we observe that the *MMDGP-nystrom* quickly converges to the optimum while the *uGP* hobbles with a larger variance. Also, as mentioned in the paper, the *skl* and *ERBF* fail to locate the robust optimum due to the mismatching of their assumptions and true input uncertainty.
>
> We also re-evaluate on the robot pushing problem with a multi-modal Gaussian mixture distribution. Each method is tested for 48 trials and the robust regrets are summarized in Figures 9c and 9d. We can observe our algorithm outperforms the others in terms of both optimization efficiency and stability.
>
> ### 4. Writing
>
> > The writing has much to be improved...
>
> Thanks for pointing them out and sorry for such a rough presentation. We have revised them accordingly and performed careful proofreading for the updated version.
>
>
> [1] Andreas Christmann and Ingo Steinwart. “Universal Kernels on Non-Standard Input Spaces”. NIPS. 2010
>
> [2] Oliveira, Rafael, Lionel Ott, and Fabio Ramos. “Bayesian Optimisation under Uncertain Inputs.” PMLR, 2019.
>
> [^a] For simplicity, all the methods in this work use an acquisition of upper confidence bound.

---

> > ### Comment · Reviewer_UjwV · 2023-08-11
> > **Additional concern**
> >
> > Thank you for your response, most of my previously raised concerns are decently addressed. I have increased my score to reflect this.
> >
> > I have one more qualm regarding the comparison of your kernel to that developed in Oliveira et. al. (2019), and to the kernels developed in "Learning from Distributions via Support Measure Machines" by Muandet et. al. (2012) and "Universal Kernels on Non-Standard Input Spaces" by Christmann and Steinwart (2010). The kernel developed in Oliveira et. al. (2019) has the form $k(P, Q) = \langle \psi_P, \psi_Q \rangle_{\mathcal H}$, and they state below this definition that "Besides the linear kernel in Equation 9, many other kernels on $\mathcal P$ can be defined via $\psi$, e.g. radial kernels using $\lVert \psi_P - \psi_Q\lVert_{H}$ as a metric on $\mathcal P$ (Muandet et al., 2012). " Since your kernel is defined as $k(P, Q) = \exp(-\alpha \lVert \psi_P - \psi_Q\lVert_{H})$ this seems to be the same as what was suggested. Muandet et. al. (2012) seems to have tested this also as a "Level 2 RBF", and seems to be very similar to that developed in Christmann and Steinwart (2010) Lemma 2.3. My concerns concretely are the following:
> >
> > 1. The fact that the proposed kernel is not new must be clearly stated and these links to previous works must be clearly discussed. On reading the paper in its current form, one has the impression that the proposed kernel is part of the work's novel contribution.
> >
> > 2. The work of Muandet et. al. (2012) suggests that there is an entire family of kernels over probability measures, of which the one in Oliveira et. al. (2019) and yours are special cases. The Nystrom approximation will be able to decrease the computation time for all of them, since they all rely on inner products between sums of kernel matrix entries and therefore all rely on the kernel matrix. I believe that this work can be made significantly more general by not focusing on the case of RBF with MMD as a metric, and since Theorem 1 does not rely on this specific kernel anyway. Or is there a good reason that there is a particular focus on this specific kernel?

---

> > > ### Author Response · Authors · 2023-08-12
> > > **Response to kernel design**
> > >
> > > We appreciate the reviewer's thorough understanding of this paper and insightful comments!
> > >
> > > + We come to this MMD-based kernel design from an intuition that, if we can find a good metric that can characterize the distance btw probability measures with a rich expressive power, we should be able to devise a robust GP based on it and guide the search for robust optimum.
> > >
> > > + This intuition turns us into a family of Integral Probabilistic Metrics (IPMs). Among them, the MMD grabs our attention because of its intrinsic connection with distance measurement in RKHS and high coincidence with the kernel trick (ref. Section 3.1 of our paper), which eventually leads us all the way to the current idea.
> > >
> > > + Given two distributions P and Q, the MMD [1] is defined as:$MMD(P, Q) = \\sup\_{f \\in \\mathcal{F}} \\big( \\mathbb{E}\_{x}[f(x)] - \\mathbb{E}_{y}[f(y)] \\big)$. By setting the function class $\mathcal{F}$ to a unit ball in an RKHS $\mathcal{H}$, the MMD can be expressed as a distance btw the mean embeddings in RKHS. In this case, our kernel becomes $k(P, Q)=\exp{(-\alpha \Vert \psi_P -\psi_Q \Vert_H)}$, which converges to the kernel family in [2] (The uGP kernel in [4] is a specialization of [2]).
> > >
> > > + Though our theoretical analysis is developed upon the RKHS, **MMD is not limited to RKHS and our kernel can go beyond the kernel family in [2]**. In fact, any function class $\mathcal{F}$ that comes with uniform convergence guarantees and is sufficiently rich can be used, which gives different expressions of MMD [1]. For example, MMD converges to the Kolmogorov metric if $\mathcal{F}$ is a class of bounded variations. Also, MMD can be expressed as other Earth-mover distances, say Wasserstein distance, with proper choice of function class $\mathcal{F}$. In this field, there are some active researches, e.g., [5, 6, 7], to design novel kernels with these IPM metrics.
> > >
> > > **In the updated manuscript, we will clarify this point clearly and discuss the connections with these existing works as the reviewer suggested.**
> > >
> > > Again, thanks for the constructive suggestion on the generalization direction. As pointed out by the reviewer, our theoretical analysis in theorem 1 does not rely on an assumption of the kernel in its current form and can be generalized. As the main scope of this paper is a practical method for robust Bayesian optimization, we will **add a discussion for this generalization direction and leave the comprehensive exploration in future work.**
> > >
> > >
> > > **Reference**
> > >
> > > [1] Gretton, Arthur, et al. "A kernel two-sample test." *The Journal of Machine Learning Research* 13.1 (2012): 723-773.
> > >
> > > [2] Muandet, Krikamol, et al. "Learning from distributions via support measure machines." *Advances in neural information processing systems* 25 (2012).
> > >
> > > [3] Christmann, Andreas, and Ingo Steinwart. "Universal kernels on non-standard input spaces." *Advances in neural information processing systems* 23 (2010).
> > >
> > > [4] Oliveira, Rafael, Lionel Ott, and Fabio Ramos. "Bayesian optimization under uncertain inputs." *The 22nd international conference on artificial intelligence and statistics*. PMLR, 2019.
> > >
> > > [5] Carriere, Mathieu, Marco Cuturi, and Steve Oudot. "Sliced Wasserstein kernel for persistence diagrams." *International conference on machine learning*. PMLR, 2017.
> > >
> > > [6] Oh, Jung Hun, et al. "Kernel wasserstein distance." *arXiv preprint arXiv:1905.09314* (2019).
> > >
> > > [7] De Plaen, Henri, Michaël Fanuel, and Johan AK Suykens. "Wasserstein exponential kernels." *2020 International Joint Conference on Neural Networks (IJCNN)*. IEEE, 2020.

---

> > > > ### Author Response · Authors · 2023-08-15
> > > > **A warm reminder on the last day of the rolling discussion**
> > > >
> > > > We kindly ask the reviewers if they have any outstanding questions or clarifications regarding our paper. We are happy to engage in a dialogue and conduct any additional requested work in the remaining discussion period. Thank you!

---

> > > > > ### Comment · Reviewer_UjwV · 2023-08-15
> > > > >
> > > > > Thank you for your response.
> > > > >
> > > > > While other distances such as the Wasserstein distance may be a special case of the MMD depending on the function class $\mathcal F$, in the context of your work, if $\mathcal F$ is not the RKHS associated with $k$ and the MMD is not a sum of kernel matrix entries, the Nystrom approximation is not applicable. Your work seems to only be applicable to the kernel family, of which uGP and the MMD in your paper are special cases. Once again, I believe that the work can be made much more general by properly discussing this fact. The empirical section ideally would then involve a few members of this kernel family, but I understand that that will take time.
> > > > >
> > > > > I believe that you will incorporate the lengthy discussion we have had into the next iteration of the paper and properly position your work in the context of the BO and kernel methods literature. I have increased my score further to reflect this.

---

> > > > > > ### Author Response · Authors · 2023-08-15
> > > > > > **Thank you for the kind review**
> > > > > >
> > > > > > We thank the reviewer again for the considered comments and in particular for suggesting the generalization direction of our idea! We will clarify these points clearly and discuss the connections to the existing works as suggested in the updated paper.

---

### Official Review · Reviewer_JQUf · 2023-07-05

**Soundness:** 3 good
**Presentation:** 3 good
**Contribution:** 2 fair
**Rating:** 5
**Confidence:** 4

**Summary:**

This work focuses on the situations where input uncertainty arises and the input values are unobservable, and introduces to measure the distance of uncertain inputs through MMD when training the Gaussian process surrogate. The authors theoretically and empirically demonstrate the effectiveness of the proposed method.

**Strengths:**

1.	The proposed method is sound and MMD can be a metric to measure the distances of uncertain inputs.

2.	The experiments are extensive to show the effectiveness of the proposed method.


**Weaknesses:**

1.	MMD-GP needs to query m times more samples than GP, which can restrict its application since BO is usually applied to the tasks where evaluating a query can be time-consuming. What about the time cost of this work compared to other methods for input uncertainty?

2.	After sampling m times for one query x, we can also get m rewards/performance. I wonder how to deal with the m rewards/performance? Line 139 shows that $D_n=${$(x_i, y_i)|x_i \sim P_i$}. However, it seems that $y_i$ also should be a distribution.

3.	Other typos:

3.1	Eq. 2 use $\xi$ to denote the noise while line 92 utilize $\epsilon$.

3.2	Line 151: “the computation and space complexities of the empirical MMD estimator scale quadratically with the sampling size N” Should N be m?

3.3	What does the training and testing samples stand for in line 162? Do you mean the number of observations and the number of queries?



**Questions:**

Please see the weakness.

**Limitations:**

No broader societal impacts are provided.

---

> ### Author Rebuttal · Authors · 2023-08-10
>
> We sincerely thank the reviewer for the insightful comments.
>
> ### 1. Sampling issue
> > MMD-GP needs to query m times more samples than GP, which can restrict its application since BO is usually applied to tasks where evaluating a query can be time-consuming. What about the time cost of this work compared to other methods for input uncertainty?
>
> Thanks for pointing this out and sorry for our obscure expression! In our problem setting, we assume our input determinant point $x_i$ is blurred by some noise $\delta(x_i)$, and then evaluate a time-consuming $f(x_i + \delta(x_i))$ to get $y_i = f(x_i + \delta(x_i)) + \xi_i$ with additional noise $\xi_i$. However, when doing MMD-based kernel evaluation, we do not have to query the time-consuming function $f$ but only assume $\delta(x_i)$ can be easily sampled, and thus we have enough samples $\\{ \delta^j(x_i) \\}_{1\le j\le m}$ for each $1 \le i \le n$ to calculate the MMD distance between $x_i + \delta(x_i)$ and $x_l + \delta(x_l)$. When calling the evaluation of $f$, we only input one sample $x_i + \delta(x_i)$, and return one evaluation $y_i$.
>
> ### 2. Misleading notations
> > After sampling m times for one query x, we can also get m rewards/performance. I wonder how to deal with the m rewards/performance. Line 139 shows that $D_n = \\{ (x_i,y_i) \vert x_i \sim P_i\\}$. However, it seems that $y_i$
>  also should be a distribution.
>
> Thanks for pointing this out and sorry for our obscure expression! This question can also be solved by our last answer, as in each query, we only input one $x_i$ to the function $f$ and thus generate one $y_i$ . Also there are some typo in the definition of $D_n$ in our origina paper: $D_n = \\{ (\\hat x_i,y_i) \vert \\hat x_i \sim P_{x_i}\\}$, here $\hat x_i$ is one random evaluation of $P_{x_i}$ when we intend to input $x_i$ into the function $f$.
>
> ### 3. Typos
> > Other typos
>
> Thanks for pointing this out, we have modified all the mentioned typos in the updated paper.

---

> > ### Author Response · Authors · 2023-08-15
> > **A warm reminder in last day of rolling discussion**
> >
> > We kindly ask the reviewers if they have any outstanding questions or clarifications regarding our paper. We are happy to engage in a dialogue and conduct any additional requested work in the remaining discussion period. Thank you!

---

> ### Comment · Reviewer_JQUf · 2023-08-21
>
> Thanks for your response and I have read the rebuttal. The authors resolve my concerns and I would like to maintain the rating.

---

### Official Review · Reviewer_Yh2J · 2023-07-06

**Soundness:** 2 fair
**Presentation:** 3 good
**Contribution:** 2 fair
**Rating:** 6
**Confidence:** 4

**Summary:**

The paper proposes a novel Bayesian Optimization (BO) algorithm that explicitly tackles input uncertainty by introducing a new integral probabilistic metric (IPM)-based kernel. The algorithm also utilizes an efficient and stable Nystrom estimator to approximate the Maximum Mean Discrepancy (MMD), which serves as the adopted IPM. Furthermore, the paper extends the GP-UCB framework by incorporating the proposed kernel and derives the corresponding upper bound for the cumulative regret. The empirical study illustrates the effectiveness of the proposed approximation and the BO algorithm.

**Strengths:**

1. The paper is generally well-organized and illustrative with figures and tabular results.
2. The theoretical result extends the existing UCB work with reasonable approximation to tackle the input uncertainty.
3. The limitation of the proposed AIRBO on dealing with high-dimensional input and lack of discussion over other IPMs is clearly stated.

**Weaknesses:**

1. The theoretical bound is arguably sound. Due to the numerical approximation of the Maximum Mean Discrepancy (MMD), it may potentially result in pseudo-metric or even worse outcomes in practice. Therefore, it remains unclear whether the upper bound of the maximum information gain $\gamma_T$ still holds and guarantees that the UCB cumulative regret is sublinear.

2. The essential justification for the proposed Nystrom estimator lies in assumption 1. While the author claims in lines 210 and 211 that the approximation error can be fairly small, the paper lacks sufficient discussion or substantiation to support this claim.

3. The empirical results are limited. The regret curve lacks statistical significance, and more extensive empirical studies on real-world applications would be desirable.

**Questions:**

1. What is SKL-UCB mentioned in line 262? Is it referring to the algorithm discussed in line 45 from [19]?
2. What is the corresponding reference for Oliveira et al. mentioned in line 50?
3. Despite the explicit treatment of input uncertainty, it is unclear to me whether tolerating the input uncertainty by fixing a larger output noise level could provide a simpler solution. The author claims that the GP model overfits (line 222) and justifies the use of MMD-GP. However, one would expect that fixing a larger noise level for the GP model would help mitigate it as well. Additionally, the existing method addressing heteroscedastic noise proposed in [1] potentially aids in mitigating the input uncertainty purely on the output side. It would be valuable to hear the authors' comments on this.
4. Could the author provide more intermediate quantitative evidence showing the effectiveness of the proposed MMD-based kernel? For example, any evidence showing that the uncertainty quantification is better than the naïve GP model that doesn't explicitly deal with the input uncertainty?

**Limitations:**

LImitations are mentioned in the comments above.

---

> ### Author Rebuttal · Authors · 2023-08-07
>
> We sincerely thank the reviewer for the constructive comments.
>
> ### Theoretical bound
> > The theoretical bound is arguably sound. Due to the numerical approximation of the Maximum Mean Discrepancy (MMD), it may potentially result in pseudo-metric or even worse outcomes in practice. Therefore, it remains unclear whether the upper bound of the maximum information gain $\gamma_T$ still holds and guarantees that the UCB cumulative regret is sublinear.
>
> We modify the expression of our key result, showing that the cumulate regret is bounded by the approximating error $e_\varepsilon$, and the maximum information gain $\hat\gamma_T$ of the **exact kernel $\hat k$**, thus no need to calculate the approximate information gain. One can check Theorem 1 in our general rebuttal part. We also discuss the order of $\hat\gamma_T$ in Theorem 2, showing that in mild assumption, $\hat\gamma_T$ satisfies an order depending on its differentiability and dimension, and thus the UCB cumulative regret is sublinear.
>
> The modified theorem 1 can be checked in the global rebuttal.
>
>
> ### Nystrom errors
> > The essential justification for the proposed Nystrom estimator lies in assumption 1. While the author claims in lines 210 and 211 that the approximation error can be fairly small, the paper lacks sufficient discussion or substantiation to support this claim.
>
> We have updated assumption 1 and added a remark after assumption 1 to discuss the general approximation error of the Nystrom estimator. Please refer to the global rebuttal.
>
> ### Experiments
> > The empirical results are limited. The regret curve lacks statistical significance, and more extensive empirical studies on real-world applications would be desirable.
>
> Thanks for pointing out this issue. We have rerun all the experiments with more trials to provide a statistically-meaningful result: Figure 9 in the attached supplementary_results.pdf reports all the updated experiment results. In particular, a performance comparison on the RKHS function with Gaussian input uncertainty is reported in Figure 9a, the result on the 1D double-peak function with beta inputs is presented in Figure 9b, while 9c and 9d compare the performance on a real-world robot pushing problem.
>
> In addition, we also add an HD optimization problem with a 10-dimensional bumped bowl problem from [2] and set the input uncertainty as a circular distribution. Figure 7b in the supplementary results shows our method yields the best performance and can locate the robust optimum in high dimension efficiently and stably.
>
>
> ### Questions
> > 1. What is SKL-UCB mentioned in line 262? Is it referring to the algorithm discussed in line 45 from [19]?
>
> Yes, the SKL-UCB employs a GP surrogate equipped with a symmetric KL-based kernel, as described in[19], to model the uncertain inputs. We have updated the citation in Iine 262.
>
>
> > 2. What is the corresponding reference for Oliveira et al. mentioned in line 50?
>
> Sorry for the mistake. We have added the correct reference to [1] in the updated paper.
>
>
> > 3. Despite the explicit treatment of input uncertainty, it is unclear to me whether tolerating the input uncertainty by fixing a larger output noise level could provide a simpler solution. The author claims that the GP model overfits (line 222) and justifies the use of MMD-GP. However, one would expect that fixing a larger noise level for the GP model would help mitigate it as well. Additionally, the existing method addressing heteroscedastic noise proposed in [1] potentially aids in mitigating the input uncertainty purely on the output side. It would be valuable to hear the authors' comments on this.
>
> This is an interesting viewpoint. In our understanding, explicit modeling of the input uncertainty not only helps to distinguish the input perturbation from other sources of randomness (e.g., measure noise) but also enables us to predict the **expected** function value under such an input uncertainty, which then can be used to guide the search for the robust optimum. This is quite reasonable and intuitive in our problem setting. On the contrary, simply setting a larger output noise to absorb the input uncertainty may mix the different sources of randomness together, rendering the output noise level hard to learn.
>
>
> > 4. Could the author provide more intermediate quantitative evidence showing the effectiveness of the proposed MMD-based kernel? For example, any evidence showing that the uncertainty quantification is better than the naïve GP model that doesn't explicitly deal with the input uncertainty?
>
> Thanks for this constructive comments. We have conducted a new experiment for modeling the uncertainty with diff.
> In this test, we intentionally design the input uncertainty to be a "step-changing" Chi-squared distribution, whose degrees of freedom parameter $df$ is 0.5 when $x \in [0.0, 0.6)$ and suddenly changes to 7.0 if $x \in [0.6, 1]$. Figure 8 in the supplementary results visualizes the uncertainty quantification of different surrogate models (The numbers followed by the surrogate name are the sampling sizes, *e.g.*, *MMDGP-nystrom(160/10)* is our method with a sampling size of 160 and sub-sampling size of 10).
>
> We can clearly observe that i) *MMDGP-nystrom* can comprehensively model the input uncertainty, evidenced by the abrupt change of its posterior distribution at the location $x=0.6$. ii) uGP with a small sampling size fails to quantify the uncertainty but a large sampling size with a higher computation cost help alleviate this problem. iii) A naïve GP model that does not explicitly deal with the input uncertainty cannot be aware of this uncertainty change at all.
>
> ### Reference
> [1] Oliveira, Rafael, Lionel Ott, and Fabio Ramos. “Bayesian Optimisation under Uncertain Inputs.” PMLR, 2019.
>
> [2] Sanders, Nicholas D., Richard M. Everson, Jonathan E. Fieldsend, and Alma A. M. Rahat. “Bayesian Search for Robust Optima.” arXiv, 2021.

---

> > ### Comment · Reviewer_Yh2J · 2023-08-15
> >
> > I appreciate the authors' detailed response to the concerns regarding the soundness of the theoretical and empirical results. I believe the new results could help alleviate the issues and enhance the overall presentation. However, the new results in Figure 8 demonstrate an improvement in uncertainty quantification but do not seem to contribute significantly to the optimization, especially when compared to uGP. Additionally, the results in Figure 9 still lack significance. I still have some reservations about the contribution of explicitly learning the input uncertainty for general optimization. Nonetheless, I appreciate the general contribution of this paper within this context.

---

> > > ### Author Response · Authors · 2023-08-15
> > > **Additional explanation**
> > >
> > > Thanks for your feedback. Here we would like to explain a little bit more about the experiment setting and results, hoping it can help dispel some concerns.
> > >
> > > In Figure 8, we aim to compare the modeling performances of different surrogates given the same set of training points. We design the input distribution to be a "step-changing" Chi-square distribution, whose $df = 0.5$ if  $x\in[0.0, 0.6)$)  and $df$ changes to 7.0 when $x\in[0.6, 1.0)$ [1]. Due to this sudden change, the uncertainty at point $x=0.6$ is expected to be asymmetric: i) as the Chi-square distribution is quite lean and sharp when $df=0.5$, the distance from $x=0.6$ to its LHS points, i.e., $x_{lhs} \in[0.0, 0.6)$ are relatively large thus their covariances are small, resulting a fast-growing uncertainty. Meanwhile, ii) when $x\geq 0.6$, the $df$ changes to 7.0, rendering the input distribution a quite flat one with a long tail at RHS. Therefore, the distances btw $x$ and its RHS points become relatively small, which leads to large covariances and small uncertainties for points in $[0.6, 1.0]$. As a result, we expect to observe an asymmetric posterior uncertainty at $x=0.6$.
> > >
> > > We have employed several surrogates in this test, including:
> > >
> > > + *MMDGP-nystrom(160/10)* is our method with a sampling size $m=160$ and sub-sampling size $h=10$. Its complexity is $O(MNmh)$, where $M$ and $N$ are the sizes of training and test samples (Note: all the models in this experiment use the same training and testing samples for a fair comparison).
> > > + *uGP(40)* is the surrogate from [2] and employs an integral kernel with sampling size $m=40$. Due to its $O(MNm^2)$ complexity, we set the sampling size $m=40$ to ensure a similar complexity as the *MMDGP-nystrom(160/10)* .
> > > + *uGP(160)* is also the surrogate from [2] but uses a much larger sampling size ($m = 160$). Given the same training and testing samples, its complexity is 16 times higher than our *MMDGP-nystrom(160/10)*.
> > > + *skl* is another robust GP surrogate equipped with a symmetric KL-based kernel, as described in[13].
> > > + *ERBF* employs an expected RBF kernel from [4] (Both *skl* and *ERBF* are designed for Gaussian inputs).
> > > + *GP* utilizes a noisy Gaussian Process model with a learnable output noise level.
> > >
> > > According to Fig 8, we observe that:
> > > + **our method, MMDGP-nystrom(160/10), can comprehensively quantify the sudden change of the input uncertainty**, evidenced by its abrupt posterior change at $x=0.6$.
> > >
> > > + However, **the *uGP (40)* with the same complexity fails to model the uncertainty correctly**. We suspect this is because *uGP* requires much larger samples to stabilize its estimation of the integral kernel and thus can perform poorly with insufficient sample size, so we also evaluate the *uGP(160)* with much higher complexity (sampling size $m=160$) and it does successfully alleviate this issue.
> > >
> > > + Apart from this, the **noisy GP model with a learnable output noise level are not aware of this uncertainty change at all**, this may be because it treats the inputs as the exact values instead of random variables.
> > >
> > > This test is only designed to examine the uncertainty quantification, but not their optimization performance. We further compare the optimization performances on the 1D synthetic functions and a real-world problem in Figure 9, as well as a 10D problem under circular distribution in Figure 7b. According to these tests, we found that:
> > >
> > > + In all experiments, **the noisy GP with learnable output noise level fails to locate the robust optimum and can get stuck at a local optimum**.
> > >
> > > + In problems with complex distributions and high dimensions, **our method outperforms the others and works quite, while the *uGP* at the same computation cost suffers the instability caused by insufficient sampling and stumbles over iterations**. This observation can be supported by the mean and standard deviation of regret in Figures 7b and 9c.
> > >
> > > Moreover, the discussion on the benefits of modeling the input uncertainty explicitly can be founded in these works: [2, 3, 4, 5] (which are already cited in our manuscript).
> > >
> > > **Reference**
> > >
> > > [1] Chi-squared distribution, Wikipedia, https://en.wikipedia.org/wiki/Chi-squared_distribution
> > >
> > > [2] Oliveira, Rafael, Lionel Ott, and Fabio Ramos. "Bayesian optimization under uncertain inputs." The 22nd international conference on artificial intelligence and statistics. PMLR, 2019
> > >
> > > [3] Moreno, Pedro, Purdy Ho, and Nuno Vasconcelos. "A Kullback-Leibler divergence-based kernel for SVM classification in multimedia applications." *Advances in neural information processing systems* 16 (2003).
> > >
> > > [4] Dallaire, Patrick, Camille Besse, and Brahim Chaib-Draa. "An approximate inference with Gaussian process to latent functions from uncertain data." *Neurocomputing* 74.11 (2011): 1945-1955.
> > >
> > > [5] Beland, Justin J, and Prasanth B Nair. “Bayesian Optimization Under Uncertainty.” In *Advances in Neural Information Processing Systems*, 5, 2017.

---

> > > > ### Comment · Reviewer_Yh2J · 2023-08-15
> > > >
> > > > Thank you for the swift response and detailed elaboration, which has further clarified the context of the results presented in Figure 8. I acknowledge the notion that an enhancement in uncertainty quantification is demonstrated around x=0.6, and there exists a series of works supporting the effectiveness of broader explicit input uncertainty treatments in optimization.
> > > >
> > > > What I would kindly request the authors to have second thoughts on is that the figures in the second row of Figure 8, illustrating the acquisition function, do not directly convey the concept that the proposed input uncertainty treatment better benefits the optimization process specifically. While I don't fully dismiss the improvements shown in Figure 9, focusing solely on Figure 8 leads me to contend that its construction should be revised to more clearly and directly reflect the advancements in Bayesian optimization-oriented uncertainty quantification.
> > > >
> > > > I hold the belief that with a more extended period of revision, it is possible that the author can orchestrate a scenario that better conveys the underlying concept that the proposed method facilitates the optimization and effectively motivates the algorithm in a similar format. The presentation would be much stronger with the improved illustration.

---

> > > > > ### Author Response · Authors · 2023-08-15
> > > > >
> > > > > Thanks a lot for the constructive and kind suggestions. We understand current experiment setup in Figure 8 can be further improved to better illustrate our motivation and will update it in the revision as the reviewer suggested.

---

### Official Review · Reviewer_jZyz · 2023-07-08

**Soundness:** 3 good
**Presentation:** 3 good
**Contribution:** 2 fair
**Rating:** 6
**Confidence:** 4

**Summary:**

The paper tackles the problem of robust Bayesian Optimization (BO) with uncertain inputs, i.e., the input values are deviated from the intended value before evaluation. The paper proposes a new technique, namely AIRBO (Arbitrary Input uncertainty Robust Bayesian Optimization), that can model the uncertain input and incorporate this uncertainty into the surrogate model, and thus can be used to guide the search of the objective function global optimum. The paper further proposes to use Nystrom approximation to reduce the computational cost. Theoretical analysis is conducted to guarantee the performance of the proposed technique. Experiments are conducted on some synthetic and one real-world problem to evaluate the performance of the proposed method.

**Strengths:**

+ The paper’s writing is generally clear and easy to understand. Illustration and figures are plotted nicely to explain the problem setting, as well as the property of the proposed method.
+ The proposed method of using MMD to construct a kernel that can incorporate uncertainty of the inputs seems to be interesting to me.
+ Theoretical analysis is conducted to guarantee the convergence property of the proposed method (although note that I have some concerns regarding the theoretical analysis).
+ The experiments are described very detailed. Various experiments are conducted to understand the performance of each component within the proposed method.


**Weaknesses:**

+ I think the application of this problem setting should be motivated much better as it’s unclear on the significance of the problem tackled in this paper. In the Introduction, the paper only mentions that this problem is quite common for robotics and process controls, but no references or further explanations are provided.
+ Some notations are quite confusing, making it hard for me to follow the proposed method. In Eq. (8), the paper shows how to compute MMD(P,Q) from the m samples {x_i}_1^m, {y_i}_1^m – but y is defined as the objective function value. Then how can we compute the kernel \hat{k}(P_{x_i}, P_{x_j}) in Eqs. (10) and (11)?
+ I also have some concerns with the theoretical analysis. In the theoretical analysis, there is no formal proof to prove the proposed kernel is actually a valid kernel. There is just one paragraph in Lines 170-173 briefly mentioning about this but no formal proof is given to know if the proposed kernel is valid, and in which scenarios it will be a valid kernel. Assumption 1 is too strong, it assumes that we already know that the error function e(P,Q) can be uniformly upper-bounded. In practice, can this assumption be true? How can we know it can be true? Theorem 1 also doesn’t have much meaning to me. In Theorem 1, \tilde{\beta}_n is defined with the maximum information gain \tilde{I} within the formula. This is not really a standard analysis in BO’s theory, normally \beta is defined by some constants. Is this maximum information gain \tilde{I} bounded?
+ Experiments are mostly conducted with synthetic functions. There is only one real-world problem being used in the experiment, and it’s quite simple to me. It’s just a 3-dim problem.
+ What is the time cost of the proposed method?


**Questions:**

Please answer my comments and questions in the Weaknesses section.

**Limitations:**

I can’t find any dedicated section that describes about the limitations of the proposed technique. There are some future work mentioned in the Conclusion, but there are no limitations mentioned there.

---

> ### Author Rebuttal · Authors · 2023-08-07
>
> We sincerely thank the reviewer for the constructive comments.  **The reference is listed in the global rebuttal**.
>
> ### 1. Motivation
> > I think the application of this problem setting should be motivated much better as it’s unclear on the significance of the problem tackled in this paper.
>
> Thanks for bringing it up. This paper aims to tackle a robust optimization problem in which the design parameters are perturbed randomly before the evaluation. Such perturbations are the manifestation of uncertainties in the design process and can arise for several reasons: *e.g*., execution noise during the control process or machining error of manufacturing. The drone measurement in Section 1 gives an example of execution noise, yet some other motivating applications include:  robot exploration [7], robot grasping [3], semiconductor design [4], and more applications can be found in [6].
>
> ### 2. Notations
> > In Eq. (8), the paper shows how to compute MMD(P, Q) from the m samples ${x_i}^m, {y_i}^m$ – but y is defined as the objective function value. Then how can we compute the kernel $\hat{k}(P{x_i}, P{x_j}$) in Eqs. (10) and (11)?
>
> Sorry for the abused notation. In Eq.8, the $y$ is not the objective function value, we use $x$ and $y$ to represent the samples from the input distributions $P$ and $Q$ respectively.  The updated Eq.8 should be:
>
> Suppose we can get $m$ samples, $\\{ x_i^{(u)} \\}\_{u=1}^m$ and $ \\{x_j^{(v)} \\}\_{v=1}^m$, from input distribution $P_{x_i}$ and $P_{x_j}$, the MMD can be empirically estimated via:
> $$
> \text{MMD}^2(P_{x_i}, P_{x_j}) \approx \frac{1}{m(m-1)} \sum_{1\leq u, v\leq m, u\neq v} \big( k(x_i^{(u)}, x_i^{(v)}) + k(x_j^{(u)}, x_j^{(v)}) \big) - \frac{2}{m^2} \sum_{1\leq u,v \leq m} k(x_i^{(u)}, x_j^{(v)}),
> $$
> where $u$ and $v$ are the sample indices and $x_i^{(u)}$ represents the $u$-th sample from $P_{x_i}$, which are consistent with the notations used in Eqs 9, 10, and 11.
>
> ### 3. Theoretical analysis
> > In the theoretical analysis, there is no formal proof to prove the proposed kernel is actually a valid kernel.
>
> For the theoretical analysis, we complement the evidence that our proposed kernel is a valid kernel: by Theorem 2.2 in [11], any type of bi-functions defined on $\mathcal{P}\times\mathcal{P}$ as $\hat k(P,Q) = \sum_{i=1}^{\infty} a_i  \langle \psi_P, \psi_Q \rangle_k$ with $a_i \ge 0$, $\psi_P,\psi_Q$ is the kernel mean map from $\mathcal{P}$ to $\mathcal{H}_k$, then $\hat k(P,Q)$ is a valid kernel; In addition if $a_i > 0$, this kernel is universal over $C(\mathcal{P})$, given the mean map $\psi$ is injective and the space of $P, Q$ defined on is compact. Thus, the Gaussian RBF kernel $\hat k(P,Q) = \exp(-\alpha \Vert \psi_P - \psi_Q \Vert_k^2)$ is ensured to be valid and universal.
>
> > Assumption 1 is too strong, it assumes...can this assumption be true? How can we know it can be true?
>
> For the concern on assumption 1, we admit that the original assumption is too strong, and we weaken the uniform condition, only assuming that each evaluation of $e(P,Q)$ has a bound with probability $1-\varepsilon$. The modified assumption 1 is stated in the global rebuttal.
>
> Note that this assumption is standard in our case: we may assume $\max_{x \in \mathcal{X}} \Vert \phi \Vert_k \le K $, where $\phi$ is the feature map corresponding to the $k$. Then when using an empirical estimator, the error between $\text{MMD}_{\text{empirical}}$ and $\text{MMD}$ is controlled by $4K\sqrt{2\log(6/\varepsilon)m^{-1}}$ with probability at least $1- \varepsilon$ according to Lemma E.1, in [5]. When using the Nystrom estimator, the error has a similar form as the empirical one, and under mild conditions, when $h = O(\sqrt{m}\log(m))$, we get the error of the order $O(m^{-{1/2}}\log(1/\varepsilon))$ with probability at least $1-\varepsilon$.
>
> Under this modified assumption, the result for our theorem 1 is also slightly modified, and we state it in the general Author Rebuttal page. We may see that the final regret is bounded in a probability $1 - \delta - n\varepsilon$ compared to the original $1 - \delta - \varepsilon$, but it matters a little: as the empirical estimator and Nystrom estimator have errors that are log-scale on $\varepsilon$, we can take $n\varepsilon$ small enough without hurting the error order.
>
> > Theorem 1 also doesn’t have much meaning to me. In Theorem 1, \tilde{\beta}_n is defined with the maximum information gain \tilde{I} within the formula. This is not really a standard analysis in BO’s theory, normally \beta is defined by some constants.
>
> For theorem 1, the $\tilde{\beta}_n$ is defined with the maximum information gain $\hat\gamma_n$ because we consider in a general case: for arbitrary input f with bounded RKHS norm, not assuming $f$ is sampled from GP process. One may checkin Theorem 2, [10] for $f$ sampled from GP process, and Theorem 3, [10] for arbitrary $f$ in RKHS space.  for $f$ assume to be sampled from the GP process, we do not need to define $\beta_n$ with $\gamma_n$, while for arbitrary $f$ in RKHS space, we need it.
>
> > Is this maximum information gain \tilde{I} bounded?
>
> We add the theorem 2 in the global rebuttal to discuss the order of the maximum information gain $\gamma_n$, showing the sublinear order of the accumulative regret bound.
>
> ### 4. Experiments
> > Experiments are mostly conducted with synthetic functions. There is only one real-world problem being used in the experiment, and it’s quite simple to me. It’s just a 3-dim problem.
>
> We appreciate the reviewers' suggestion, we have added an HD optimization problem with a 10-dimensional bumped bowl problem from [8] and set the input uncertainty as a circular distribution. Figure 7b in the supplementary results shows our method yields the best performance and can locate the robust optimum in high dimension efficiently and stably.
>
> ### 5. Time
> > What is the time cost of the proposed method?
>
> Table 1 in the paper reports the time cost for our method.

---

> > ### Author Response · Authors · 2023-08-15
> > **Reminder in the last day of the Rolling Discussion**
> >
> > We kindly ask the reviewer if they have any outstanding questions or clarifications regarding our paper. We are happy to engage in a dialogue and conduct any additional requested work in the remaining discussion period. Thank you!

---

> > > ### Comment · Reviewer_jZyz · 2023-08-18
> > > **Thank you for your response**
> > >
> > > Dear authors,
> > >
> > > Thank you for your response. The response has addressed most of my concerns, so I decided to increase my score. I can't increase more because I'm not too sure regarding the setup of the optimization problem targetting in this paper. Perhaps, the authors could consider to motivate it much better.

---

> > > > ### Author Response · Authors · 2023-08-19
> > > > **Concrete motivating cases**
> > > >
> > > > We appreciate the reviewer's feedback. Here we provide two concrete examples of robust optimization, hoping they can better clarify the motivation of our paper.
> > > >
> > > > **[Case 1]  Design & Manufacture of  wireless antenna**
> > > > In wireless communication systems (e.g., 4G/WiFi), the antenna is an important device for sending & receiving electromagnetic waves. It typically consists of conductive elements, such as wires, rods, or metal surfaces. These elements are precisely shaped and positioned to determine the antenna's characteristics, *e.g.*,  the radiation pattern, polarization, frequency range, and etc. However, during its manufacturing or fabrication process, machine errors occur inevitably because of various factors, such as inaccuracies in the machining equipment, variations in material properties, and imprecise assembly techniques. These errors deviate the design parameters from the desired specifications and can have significant effects on antenna performance [1].
> > > >
> > > > For example, suppose we want to optimize the length of the antenna $x$ to maximize its gain $y$ at a frequency range of $[1.424, 2.75)$ GHz. Let the machine error be described by a probability distribution $P(x)$, the antenna gain of the final product is $f(x^* +\delta), \delta \sim P(x^*)$, where $x^*$ is the global optimum of the target function. For some antennas, the global optimum can be quite "unstable": a small perturbation in $x$ can degrade its performance dramatically (e.g., the RKHS function in Figure 4a of the paper). In these cases, instead of finding the global optimum, we aim to find a robust optimum whose average performance is best under the machine errors:
> > > > $$
> > > > x^r = \text{argmax}\_{x} \int_\delta f(x+\delta) d\_{P(x)}
> > > > $$
> > > > Moreover, as the machine errors derive from multiple random sources, its probability distribution $P(x)$ can be quite complex.
> > > >
> > > >
> > > >
> > > > **[Case 2] Robust robot grasp**
> > > >
> > > > Yet another motivating case comes from the robot grasp [2]. In this problem, we aim to control a robot hand to grasp an object. In particular, we need to decide the optimal translations and rotations of the robot hand: $\mathbf{x}=(\Delta_x, \Delta_y, \Delta_z, \theta_x, \theta_y, \theta_z)$. However, controller errors and execution noises happen during grasp and deviate hand configuration $x^{\prime} = \mathbf{x} + \mathbf{\delta}, \mathbf{\delta} \sim P(\mathbf{x}) $, preventing the precise positioning of the robot hand in the desired pose for a stable grasp. In addition, the grasp performance is often measured via a wrench space analysis [3] and its discontinuous nature sometimes renders it quite sensitive to a small configuration change, which further complicates this problem.
> > > >
> > > > As such, when planning for a robotic grasp, we must consider the impact of the uncertainty in the input space (positioning errors) to ensure a robust grasp:
> > > > $$
> > > > \mathbf{x}^r = \text{argmax}\_{x} \int_{\delta} f(x+\delta) d\_{P(x)}
> > > > $$
> > > >
> > > >
> > > >
> > > > ### Summary
> > > >
> > > > In general, both of these two cases involve the design parameters being perturbed before the evaluation by random errors and the targeting function can be quite unstable, i.e., a small change in $x$ may result in a large fluctuation in $y$. This kind of problem is quite common in the real world, say semiconductor design [4], and a more complete application list can be found in [5].
> > > >
> > > > Locating the robust optimum in these tasks can be formed as an optimization problem in the expected form and robust optimization methods can be used. Compared with the other methods, our solution can model complex input distributions comprehensively and perform the optimization efficiently.
> > > >
> > > >
> > > >
> > > > **Reference**
> > > >
> > > > [1] Stutzman, Warren L., and Gary A. Thiele. Antenna theory and design. John Wiley & Sons, 2012.
> > > >
> > > > [2] Nogueira, José, et al. "Unscented Bayesian optimization for safe robot grasping." *2016 IEEE/RSJ International Conference on Intelligent Robots and Systems (IROS)*. IEEE, 2016.
> > > >
> > > > [3] Weisz, Jonathan, and Peter K. Allen. "Pose error robust grasping from contact wrench space metrics." *2012 IEEE international conference on robotics and automation*. IEEE, 2012.
> > > >
> > > > [4] Ng, Tsan Sheng, Yang Sun, and John Fowler. "Semiconductor lot allocation using robust optimization." *European Journal of Operational Research* 205.3 (2010): 557-570.
> > > >
> > > > [5] Gabrel, Virginie, Cécile Murat, and et al. “Recent Advances in Robust Optimization: An Overview.” *European Journal of Operational Research* 235, no. 3 (June 2014): 471–83.

---

### Author Rebuttal · Authors · 2023-08-09

We sincerely thank all the reviewers for their insightful comments and constructive suggestions. In this global rebuttal, we mainly provide the updated version of our theoretical results and a discussion on the limitations, followed by an attached pdf for the supplementary experiments (The other detailed feedback can be found under each reviewer's comments).

### Theoretical Results
For the theoretical part, we summarize the amended theorems/assumptions/descriptions here.

**Assumption 1** For any $\varepsilon > 0$, $P,Q \in \mathcal{P_{\mathcal{X}}}$, we may choose an estimated $\tilde k(P,Q)$ such that the error function $e(P,Q)$ can be upper-bounded by $e_\varepsilon$ with probability at least $1-\varepsilon$, that is, $\mathbb{P}\left(|e(P,Q)| \le e_\varepsilon\right) > 1-\varepsilon.$

**Discuss on Assumption 1** Note that this assumption is standard in our case: we may assume $\max_{x \in \mathcal{X}} \Vert \phi \Vert_k \le K $, where $\phi$ is the feature map corresponding to the $k$. Then when using empirical estimator, the error between $\text{MMD}_{\text{empirical}}$ and $\text{MMD}$ is controlled by $4K\sqrt{2\log(6/\varepsilon)m^{-1}}$ with probability at least $1- \varepsilon$ according to Lemma E.1, in [5]. When using the Nystrom estimator, the error has a similar form as the empirical one, and under mild conditions, when $h = O(\sqrt{m}\log(m))$, we get the error of the order $O(m^{-{1/2}}\log(1/\varepsilon))$ with probability at least $1-\varepsilon$.


**Theorem 1** Let $\delta >0, f\in \mathcal{H}\_k,$ and the corresponding $ \Vert \hat f \Vert_{\hat k}  \le b, \max_{x\in \mathcal{X}} |f(x)| = M$. Suppose the observation noise $\zeta_i = y_i -f(x_i) $ is $\sigma_\zeta$-sub-Gaussian, and thus with high probability $|\zeta_i|< A$ for some $A>0$. Assume that both $k$ and $P_x$ satisfy the conditions for $\Delta f_{P_x}$ to be $\sigma_E$-sub-Gaussian, for a given $\sigma_E > 0$. Then, under Assumption1 with $\varepsilon >0$ and corresponding $e_{\varepsilon}$, setting $\sigma^2 = 1+\frac{2}{n}$, running Gaussian Process with acquisition function
$\\tilde \\alpha(x|\\mathcal{D}_n) = \\tilde\\mu_n(P_x)  + \\beta_n  \\tilde\\sigma_n(P_x),$ where
$\\beta_n =b + \\sqrt{\\sigma_E^2 + \\sigma\_\\zeta^2}\\sqrt{2(\\hat \\gamma\_n + 1 - \\ln\\delta)  },$
  we have that the uncertain-inputs cumulative regret satisfies:

$\\tilde R_n \\in O ( \sqrt{n\hat\gamma_n(\hat\gamma_n - \ln \delta)}  + n^2\sqrt{(\hat\gamma_n - \ln \delta) e_\varepsilon}  + n^3 e_\varepsilon )$
 with probability at least $1-\delta - n\varepsilon$. Here $\tilde R_n = \sum_{t=1}^n \tilde r_t$, and $\\tilde r\_t = \\max\_{x\\in \\mathcal{X}} \\mathbb{E}\_{P\_x}[f] - \\mathbb{E}\_{P\_{x\_t}} [f]$


**Theorem 2** Suppose $k$ is $r$-th differentiable with bounded derivatives and translation invariant, i.e., $k(x,y) = k(x-y,0)$.  Suppose the input uncertainty is i.i.d., that is, the noised input density satisfies $P_{x_i}(x) = P_0(x - x_i), \forall x_i \in \mathcal{X}$. Then if the space $\mathcal{X}$ is compact in $\mathbb{R}^d$, the maximum information gain $\hat\gamma_n$ satisfies
\begin{equation}
    \hat\gamma_n = O(n^{(d^2 +d)/(r+d^2+d)} \log(n)).
\end{equation}
Thus, when $r > d(d+1)$, the accumulated regret is sub-linear with respect to $n$ assuming $e_\varepsilon$ is small enough.

### Limitations
Here we provide a discussion of the limitation & further work:
1. The choices of sampling and subsampling size are highly related to the distribution type, dimension, and rank of the kernel matrix. For now, we set these parameters heuristically via experiments, it is worth devising a learnable way to determine their best values automatically.
2. As the power of MMD-based two-sample tests drops polynomially with problem dimension [9], our current method can only alleviate this issue with a quadratic computation cost, which hinders a scale to very high-dimensional problems. We leave the exploration to HD problems for future work.

### Reference

[1] Andreas Christmann and Ingo Steinwart. Universal Kernels on Non-Standard Input Spaces. In:NeurIPS, 2010

[2] Srinivas N, Krause A, Kakade S M, et al. Gaussian process optimization in the bandit setting: No regret and experimental design[J]. arXiv, 2009.

[3] Nogueira, Jose, Ruben Martinez-Cantin, et al. “Unscented Bayesian Optimization for Safe Robot Grasping.” In IROS, 2016.

[4] Tsan Sheng Ng, Yang Sun, et al.,Semiconductor lot allocation using robust optimization, European Journal of Operational Research, Volume 205, Issue 3, 2010.

[5] Chatalic, Antoine et al. Nyström Kernel Mean Embeddings. ICML 2022.

[6] Gabrel, Virginie, Cécile Murat, and et al. “Recent Advances in Robust Optimization: An Overview.” *European Journal of Operational Research* 235, no. 3 (June 2014): 471–83.

[7] Oliveira, Rafael, Lionel Ott, and et al., “Bayesian Optimisation under Uncertain Inputs.” AISTATS, 2019.

[8] Sanders, Nicholas D., Richard M. Everson, et al., Bayesian Search for Robust Optima. arXiv, 2021.

[9] Ramdas, Aaditya et al. “On the Decreasing Power of Kernel and Distance Based Nonparametric Hypothesis Tests in High Dimensions.” AAAI Conference on Artificial Intelligence (2014).

[10] Srinivas N, Krause A, Kakade S M, et al. Gaussian process optimization in the bandit setting: No regret and experimental design[J]. arXiv preprint arXiv:0912.3995, 2009.

[11] Christmann A, Steinwart I. Universal kernels on non-standard input spaces[J]. Advances in neural information processing systems, 2010, 23.

---

### Decision · Program_Chairs · 2023-09-21

**Decision:**

Accept (poster)

**Comment:**

There were initially some non-minor concerns, particularly around the kernel validity and restrictiveness of the main assumption.  The concerns mostly seem to have been alleviated following the rebuttal.  However, the paper might be approaching the boundary of what amount of required editing is acceptable for camera-ready.  The authors should very carefully incorporate the required changes, including some of the writing clarity as mentioned by Reviewer UjwV.

In addition to the reviewer comments, I would mention some weaknesses in the main theoretical result.  The weakness in the sample size (e.g., n^6 scaling) is already discussed following the theorem, which is appreciated.  Another weakness is having gamma_n * sqrt(n) dependence – in the vanilla setting, several algorithms are now known getting the much better sqrt(n * gamma_n) dependence instead:
- Finite-time analysis of kernelised contextual bandits
- A domain-shrinking based Bayesian optimization algorithm with order-optimal regret performance
- High-dimensional experimental design and kernel bandits
- Gaussian process bandit optimization with few batches
Since the authors are tackling a more challenging setting and have a relatively practical focus, I wouldn’t reject based on this limitation, but I do suggest discussing it.